# Model–Behavior Alignment under Flexible Evaluation: When the Best-Fitting Model Isn't the Right One

**Itamar Avitan**[1,2,3]     **Tal Golan**[1,2,3]
[1]Department of Industrial Engineering and Management
[2]Data Science Research Center
[3]School of Brain Sciences and Cognition
Ben-Gurion University of the Negev
avitanit@post.bgu.ac.il
golan.neuro@bgu.ac.il

## Abstract

Linearly transforming stimulus representations of deep neural networks yields high-performing models of behavioral and neural responses to complex stimuli. But does the test accuracy of such predictions identify genuine representational alignment? We addressed this question through a large-scale model-recovery study. Twenty diverse vision models were linearly aligned to 4.5 million behavioral judgments from the THINGS odd-one-out dataset and calibrated to reproduce human response variability. For each model in turn, we sampled synthetic responses from its probabilistic predictions, fitted all candidate models to the synthetic data, and tested whether the data-generating model would re-emerge as the best predictor of the simulated data. Model recovery accuracy improved with training-set size but plateaued below 80%, even at millions of simulated trials. Regression analyses linked misidentification primarily to shifts in representational geometry induced by the linear transformation, as well as to the effective dimensionality of the transformed features. These findings demonstrate that, even with massive behavioral data, overly flexible alignment metrics may fail to guide us toward artificial representations that are genuinely more human-aligned. Model comparison experiments must be designed to balance the trade-off between predictive accuracy and identifiability—ensuring that the best-fitting model is also the right one.

## 1   Introduction

The search for mechanistic explanations of human cognition, in combination with rapid advances in deep learning, has motivated the use of stimulus representations in pretrained neural networks as models of the biological representation of complex stimuli. Even without modification, activation patterns in artificial neural networks (ANNs) trained on visual tasks show surprising correspondence with cortical visual representations [1–4] and visual perceptual judgments [4–11]. When evaluation is made flexible by fitting linear weights to improve the alignment between ANN representations and brain [10, 12–14] or behavioral data [7, 15–17], this approach often achieves predictive accuracy exceeding that of any other computational model. In some neuroscientific applications (e.g., brain–computer interfaces), accurate prediction is useful regardless of the underlying mechanism. In contrast, basic-science studies in computational neuroscience often rely on the assumption that a neural network whose representations are more predictive of brain or behavioral data is a better model of the mechanisms underlying the observed biological data. For models evaluated without further data-driven fitting, high predictive accuracy occurring by chance is unlikely. However, once flexible, data-driven fitting procedures are employed, an important question arises: does predictive accuracy under flexible evaluation reflect genuinely shared representations? [18–21]. This question carries

39th Conference on Neural Information Processing Systems (NeurIPS 2025).

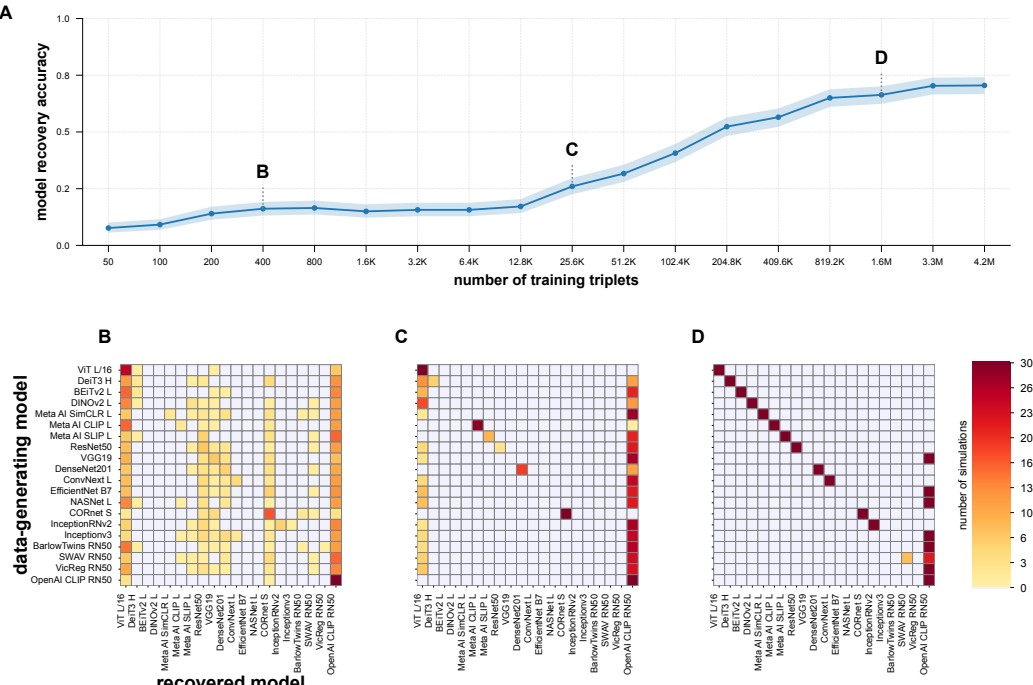

Figure 1: **(A)** Model recovery accuracy under *linear probing* with a $p \times p$ transformation matrix across different dataset sizes. **(B, C, D)** Confusion matrices at three training set sizes (400, 25.6K, and 1.6M triplets). Each matrix row corresponds to the data-generating model, and each matrix column to the recovered model. Diagonal entries represent correct model recovery. **Model recovery accuracy does not reach 80% even for millions of triplets.**

weight since there are good reasons to employ flexible evaluation. A complete yet fully emergent representational alignment is unlikely even for models whose processing is qualitatively similar to that of humans. Furthermore, inter-individual variability may also motivate flexible alignment metrics [22, 23]. And yet, flexible evaluation may incur a hidden cost: when each model is allowed to adjust to best fit brain or behavioral data, predictive accuracy—even when obtained with held-out data—may no longer serve as a meaningful index of representational alignment.

Here, we employ a *model recovery* approach to test whether predictive accuracy, as measured using current analytic methods, is indicative of the probed representations. Specifically, we use the behavioral THINGS odd-one-out dataset. We fit a linear transformation for each of a diverse set of ANN representations to predict the empirical human judgments (Fig. 2). Then, in each simulation, we sample a synthetic behavioral dataset from one of the fitted models, fit each model to the synthetic data as if it were a real experiment, and compute the models' cross-validated prediction accuracy with respect to this synthetic data. We then ask: does predictive accuracy correctly recover the data-generating model?

**Related Work.** There is growing concern about the ability of commonly used representational alignment metrics to accurately identify underlying representational structures [10, 14, 18, 19, 24–31]. The gold-standard approach for evaluating such risk is model recovery simulations, where data is simulated using the predictions of each of the models in turn to test whether the evaluation procedure identifies the data-generating model [32–35]. However, applying model recovery analyses to comparisons of ANNs as models of biological representations has largely been limited to simulations using noiseless ANN activations as observed data.

Kornblith et al. [24] compared activations from ANNs trained with different random initializations and found that flexible metrics such as linear encoding models and canonical correlation analysis failed to distinguish among specific layers, whereas an inflexible metric (centered kernel alignment, CKA) succeeded. However, since the flexible metrics were not cross-validated, their failure in the large $p$ regime could be attributed to overfitting.

Han et al. [18] compared the ability of CKA and linear regression to identify ANN architectures across initializations. When the ground-truth model was included among the candidates, CKA ranked it highest. CKA did not differentiate well between model families (convolutional networks and transformers) when the ground-truth model was held out. Cross-validated linear regression identified most—but not all—models when the ground-truth architecture was present. It was somewhat less likely than CKA to misidentify the model family when the ground-truth model was held out. However, as Han et al. [18] emphasize, these evaluations were conducted in an intentionally idealized setting: the observed data in each simulation were deterministic ANN activations, subject to no measurement noise or trial-to-trial variability.

Schütt et al. [36] demonstrated successful recovery of different cortical areas with non-flexible representational similarity analysis (RSA) using realistic, subsampled fMRI and calcium-imaging data, treating areas as alternative models. However, their validation of flexible RSA using ANN activations focused on calibration of statistical inference rather than evaluating model recovery performance in noise-calibrated settings.

Overall, the current literature does not alleviate the concern that model comparison may be inaccurate when deep neural networks are flexibly evaluated against real, noisy biological data. On real data, model comparisons are strongly constrained by signal-to-noise ratio. Specifically, without a calibrated noise model, one cannot obtain a realistic estimate of model recovery accuracy—the probability of correctly identifying the ground-truth model among the alternatives. Model recovery accuracy is a key diagnostic for model comparative procedures [35], generalizing statistical power $(1 - \beta)$ to comparisons among more than two competing alternatives.

Faithfully simulating neural data is difficult because it must capture multivariate noise and signal correlation structures. Here, we turn to simulating realistic *behavioral* experiments involving large datasets of discrete similarity judgments. Similarity judgments have long been used to infer latent spaces, most famously via multidimensional scaling (MDS) [37]. Online testing enables such experiments at scale. Specifically, Hebart et al. [38] introduced the THINGS odd-one-out dataset [39, 40], consisting of 4.7 million responses. In each trial, participants viewed three distinct images and selected the one they considered to be the odd one out. The images were randomly drawn from a standardized set of 1,854 photographs of recognizable objects [39]. The odd-one-out task is robust to differences in how participants use rating scales [38] and is straightforward to model probabilistically. Embedding-based models such as SPoSE [38] and VICE [41] capture judgments in THINGS odd-one-out with high accuracy. However, they are *non-image-computable*: they lack a forward mapping from raw pixels and thus cannot generalize to novel stimuli, and more importantly, do not explain how the representations arise. Neural-network-based models circumvent these limitations by providing fully image-computable candidate representations [1, 7, 12, 42, 43]. In a large-scale study, Muttenthaler et al. [17] used the representations of pre-trained ANNs to predict the human judgments in THINGS odd-one-out either without further fitting ("zero-shot") or under linear transformations ("linear probing"). They found that flexible evaluation substantially improved prediction performance; a CoAtNet [44] trained on image–text alignment [45] and a supervised vision transformer classifier were the top performers. This tie between functionally distinct models raises concerns that—even with massive data—predictive gains from linear probing may come at the cost of reduced identifiability.

**Our contributions:**

1. We formalize and conduct large-scale model recovery simulations for representational alignment in discrete behavioral tasks, using neural networks reweighted to mimic human judgment patterns and calibrated to match the human noise ceiling.

2. We show that—even with millions of training triplets—standard linear probing fails to reliably recover the data-generating model, with recovery plateauing below 80% accuracy (Fig. 1). This result challenges the interpretability of predictive accuracy under flexible evaluation.

3. We identify two sources of model misidentification: alignment-induced shifts in representational geometry and elevated post-alignment effective dimensionality.

4. We demonstrate that even with substantial datasets of odd-one-out judgments, there is a sharp trade-off between predictive accuracy and model identifiability. While flexible evaluation increases the apparent alignment with human responses, it can compromise model identifiability by diminishing inter-model distinctions.

## 2 Methods

### 2.1 Mapping neural network representations to human odd-one-out judgments

We followed the data-analytic approach of Muttenthaler et al. [17], with one notable modification (see *Regularization* below). For each pre-trained ANN, we extracted the final representational layer (see Appendix A.8) activation matrix $\mathbf{X} \in \mathbb{R}^{n \times p}$, where $n = 1{,}854$ denotes the number of images and $p$ the number of units. We then applied a model-specific learnable linear transformation $\mathbf{W} \in \mathbb{R}^{p \times p}$ to produce the transformed representation matrix $\mathbf{XW}$.

We computed a representational similarity matrix (RSM) $\mathbf{S} \in \mathbb{R}^{n \times n}$, where the similarity between each pair of stimuli is defined as the inner product between their transformed representations:

$$\mathbf{S} = (\mathbf{XW})(\mathbf{XW})^\top = \mathbf{XWW}^\top \mathbf{X}^\top \tag{1}$$

When $\mathbf{W}$ is set to the identity matrix, the RSM directly reflects the model's representational geometry, enabling parameter-free ("zero-shot"; [17]) evaluation of the model against human judgments. When $\mathbf{W}$ is optimized to predict human judgments, the RSM flexibly adjusts to best fit the human representational geometry ("linear-probing"; [17]).

For each trial in the human data, the similarities among the three presented images determine the model's predicted choice: the odd-one-out is the stimulus that is *not* part of the most similar pair. To obtain probabilistic predictions of odd-one-out judgments, a softmax is applied over the pairwise similarities within each triplet [17]. Given a trial with image triplet $\{a, b, c\}$ and representations of model $M$, the probability of choosing image $a$ as the odd-one-out is defined by

$$p(\text{odd-one-out} = a \mid \text{triplet} = \{a, b, c\}, M) = \frac{\exp(S_{b,c}/T)}{\exp(S_{a,b}/T) + \exp(S_{a,c}/T) + \exp(S_{b,c}/T)}. \tag{2}$$

Here, $T$ is a temperature parameter held constant during fitting and adjusted later during calibration.

To fit $\mathbf{W}$ to choice data, we used full-batch L-BFGS [46] to minimize the negative log-likelihood of the probabilistic predictions plus a regularization term:

$$\mathbf{W}^* = \arg\min_{\mathbf{W}} -\frac{1}{N_{trials}} \sum_{i=1}^{N_{trials}} \log \underbrace{p(\text{odd}-\text{one}-\text{out} = r_i \mid \{a_i, b_i, c_i\}, M)}_{\text{model prediction in trial } i} + \lambda \mathcal{R}(\mathbf{W}), \tag{3}$$

where $a_i$, $b_i$, and $c_i$ index the images presented in the i-th trial, $r_i$ the corresponding human odd-one-out choice.

**Regularization.** The Frobenius-norm-based regularization of $\mathbf{W}$, as used in [17], can degrade performance below zero-shot levels when strong penalties are applied. We replaced the regularization term with one that shrinks $\mathbf{W}$ toward a scalar matrix (see also [47]).

$$\mathcal{R}(\mathbf{W}) = \min_{\gamma} \|\mathbf{W} - \gamma \mathbf{I}\|_F^2 = \|\mathbf{W}\|_F^2 - \frac{(\text{tr}(\mathbf{W}))^2}{p}. \tag{4}$$

An analytic derivation is provided in Appendix A.1, and an empirical comparison to Frobenius-norm regularization is shown in Figure S1.

**Calibration.** While probabilistic models are often calibrated to minimize their negative log probability on held-out data [48], here we adjust the temperature parameter to ensure that the variability of simulated responses matches that of human judgments. When different participants judge the same randomly sampled triplet, their responses agree only about two-thirds of the time [38]. To reproduce this variability in simulation, we used responses from the THINGS odd-one-out noise ceiling experiment, in which 30 participants judged the same 1,000 triplets. We estimated the noise ceiling using a leave-one-subject-out procedure: for each triplet, we removed one participant's response, took the majority vote of the remaining 29, and recorded whether the held-out answer matched that vote. We then repeated this step for all participants and averaged the match rates across all triplets (see Appendix A.2.1 for mathematical formulation). For each fitted model, we tuned the temperature parameter $T$ so that, when sampling responses to these 1,000 triplets according to the model's

probabilistic predictions, the resulting prediction accuracy matched the human leave-one-subject-out noise ceiling estimate (67.8%). This estimate was computed by predicting each experimental trial using the most common response among the other participants who viewed the same triplet. See Appendix A.2.1 for implementation details. Note that, unlike standard calibration—which increases predictive entropy in less accurate models—this procedure matches each model's response variability to the level observed in human judgments, independently of its predictive accuracy.

## 2.2 Model recovery experimental setup

To evaluate the identifiability of alternative neural network models of human perceptual representation, we simulated model-comparison experiments in which behavioral data—normally collected from humans—was replaced with simulated responses generated by one of the models. Within a simulation, synthetic behavioral data were compared to the predictions of each of the candidate models using flexible model-behavior evaluation (i.e., linear probing). Suppose the widely employed analytic approach of linearly transforming neural network representations is valid. In that case, the specific model that has generated the data in a simulation should achieve the highest predictive accuracy, thereby supporting correct model recovery (see Fig. S2 for a visual illustration of the process). This setup is analogous to data distillation: the candidate models act as "students" attempting to approximate the input-output function of a "teacher"—the data-generating model. Model recovery is successful when the best-performing student is the teacher itself, rather than an alternative model.

Importantly, real human data (THINGS odd-one-out) was used only for shaping the predictions of the data-generating models; during model comparison, each model was fitted from scratch to the synthetic data, closely emulating the constraints of real data analysis, where neither the ground-truth model nor its model-to-behavior mapping parameters are known.

**Human aligned models.** We assembled a set of 20 ANNs of diverse architectures and training tasks (model details in Table S1). To generate synthetic data under the hypothesis that neural network representations can and should be linearly transformed to match empirical behavioral data, we first aligned each model to the THINGS odd-one-out training dataset (4.5 million triplets). For each model $M$, we used three-fold cross-validation over disjoint image subsets [17] to select the optimal regularization hyperparameter $\lambda_{M \to \tau}$, then fitted a model-specific transformation matrix $\mathbf{W}_{M \to \tau}$ ($M \to \tau$ denotes mapping model $M$ to THINGS odd-one-out, $\tau$), and finally calibrated the model's temperature to reproduce the human noise ceiling (see *Calibration* above). This procedure was applied to each neural network model independently. The resulting aligned models represent each neural network's best approximation to human judgments under a linear model-to-behavior mapping assumption, and serve exclusively as data-generating models in the simulations that follow.

**Simulated model comparison.** Given a set of random triplets (sampling details in Appendix A.3), we designated one of the 20 aligned models as the *data-generating model* and used its probabilistic predictions to synthesize human responses. Specifically, the data-generating model defined a categorical distribution over the three items of each triplet, from which we sampled a single response, emulating the experimental paradigm employed by Hebart et al. [38], in which each triplet was presented to a single participant. For each simulation, this procedure resulted in sets of discrete responses for the training, validation, and test triplets.

Once the synthetic responses had been generated, all 20 models, including the data-generating model, were independently fitted to the synthetic training data *from scratch*, with $\mathbf{W}$ initialized to the identity matrix. For each model, an optimal regularization hyperparameter was selected using the validation triplets, $\mathbf{W}$ was optimized on the training triplets, and prediction accuracy was evaluated on the test triplets. Because the data-generating model was calibrated, the human noise ceiling bounded prediction accuracy from above. However, if the evaluated model's predictions diverged from those of the data-generating model, the prediction accuracy could be lower. Even the data-generating model was not guaranteed to attain the noise ceiling, since it was evaluated on the test triplets after fitting $\mathbf{W}$ to limited synthetic responses rather than the full THINGS odd-one-out dataset.

In each simulation, this procedure was repeated across the three cross-validation folds (testing generalization to new images [17]), averaging the resulting prediction accuracy. For a model recovery simulation to be successful, the mean test prediction accuracy of the model that generated the data must be the highest among all of the models.

**Model recovery accuracy.**   For each stimulus set size, we repeated the simulation using 30 different random seeds to sample the stimuli. For each seed, we treated each of the 20 models as the data-generating model in turn, yielding 600 simulations per stimulus set size. Model recovery accuracy was defined as the proportion of simulations in which the data-generating model achieved the highest cross-validated test accuracy among all candidate models. We summarize our results in a confusion matrix defined as follows (full formulation is provided in Appendix A.4):

Let $\mathcal{M} = \{M_1, M_2, \ldots, M_N\}$ be a set of $N$ models, with indices $i, j \in \{1, \ldots, N\}$. Let $M_{i \to \tau}$ mark model $M_i$ as the data-generating model after it has been fitted and calibrated on human responses $\tau$. For each simulated dataset $d \in \{1, \ldots, D\}$ we define $M_{j \to i}^{(d)}$ the candidate model $M_j$ fitted to the simulated predictions of $M_{i \to \tau}$ on simulated dataset $d$. Let $\mathrm{Acc}(M_{j \to i}^{(d)} \mid M_{i \to \tau})$ be the predictive accuracy achieved by candidate model $M_j$ on the $d$-th test set responses generated by the ground-truth model $M_{i \to \tau}$ (see definition in Eq. 10). The model confusion matrix is defined by:

$$C_{ij} = \sum_{d=1}^{D} \left[ \mathrm{Acc}(M_{j \to i}^{(d)} \mid M_{i \to \tau}) = \max_{m \in \{1, \ldots, |\mathcal{M}|\}} \mathrm{Acc}(M_{m \to i}^{(d)} \mid M_{i \to \tau}) \right], \quad C \in \mathbb{N}^{|\mathcal{M}| \times |\mathcal{M}|}. \tag{5}$$

Entry $C_{ij}$ denotes the number of simulated datasets (out of $D$) for which candidate model $M_j$ was the best predictor of data generated by model $M_i$. *Model recovery accuracy* is the empirical probability of correctly identifying the ground-truth model (chance level $= 1/|\mathcal{M}|$; perfect recovery $= 1$):

$$\text{Model Recovery Accuracy} = \sum_{i=1}^{|\mathcal{M}|} C_{ii} \left/ \sum_{i=1}^{|\mathcal{M}|} \sum_{j=1}^{|\mathcal{M}|} C_{ij} \right. . \tag{6}$$

## 3   Results

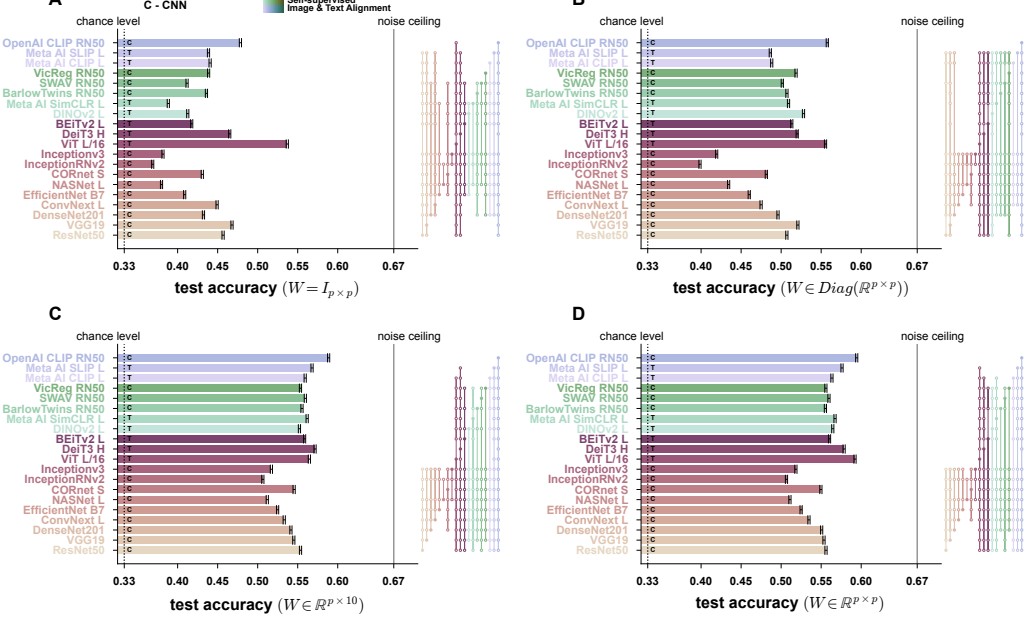

Figure 2: Model test prediction accuracy on the THINGS odd-one-out dataset across varying levels of evaluation flexibility. **(A)** Zero-shot evaluation (using each model's original embedding). **(B)** Linear probing with a diagonal transformation matrix, fitting $p$ parameters. **(C)** Linear probing with a $p \times 10$ rectangular transformation matrix. **(D)** Linear probing with a $p \times p$ full matrix.
Significance plots: a filled dot connected to an open dot indicates that the filled-dot model had significantly higher accuracy (p-value $< 0.05$, sign test, Bonferroni-corrected across 190 comparisons).

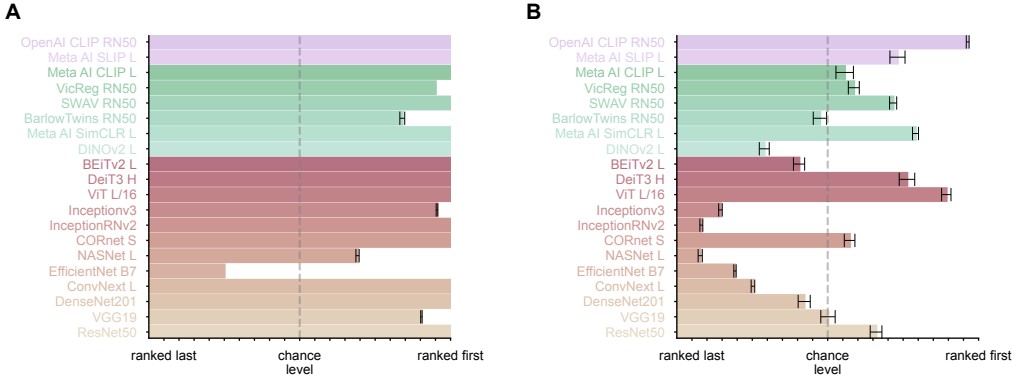

Figure 3: **(A)** Mean rank of each model's predictive accuracy when it generated the data. A mean rank above 1 indicates systematic misidentification—other models more often achieved higher predictive accuracy on data it produced. **(B)** Mean rank when the model did not generate the data. In the absence of bias, average ranks should be near chance level (dashed line). **Model misidentification is systematic—biased toward some models and away from others.** All results were computed on simulated datasets with 4.2M training triplets.

## 3.1 Comparing model predictions to empirical behavioral responses.

Before conducting the simulation study, we examined the prediction accuracy of the 20 candidate models on the THINGS odd-one-out dataset under both zero-shot and flexible evaluation settings (Fig. 2, Table S1; 3-fold cross-validation with disjoint image sets). As in Muttenthaler et al. [17], linear probing yielded higher accuracy than zero-shot evaluation. Furthermore, greater flexibility—moving from a diagonal to a rectangular to a full $\mathbf{W}$—consistently yielded additional gains.

Under the most flexible evaluation (full $\mathbf{W}$; Fig. 2D), several models achieved near–noise-ceiling predictive accuracy of human responses, with no single model performing significantly better than the others. Note that if the analytic strategy were guided solely by prediction accuracy, the most flexible evaluation would appear to be the obvious choice.

## 3.2 Model recovery simulations

**Recovery accuracy improves with data size but plateaus below 80%.** We ran simulations across 18 training set sizes (i.e., the number of synthetic triplets used to fit $\mathbf{W}$ in each fold), logarithmically spaced between 50 and 4.2 million training triplets. As described in Section 2.2, in each simulation, synthetic data were generated from a model aligned to the full THINGS odd-one-out dataset via a fitted $p \times p$ matrix, and the candidate models competed to predict the synthetic responses, each fitting a $p \times p$ matrix to the synthetic training set, and then tested on held-out synthetic responses. Model recovery accuracy as a function of training set size is shown in Figure 1A. For small datasets (i.e., those with thousands of triplets), model recovery accuracy remained below 20%. Recovery accuracy increased with dataset size; however, even with 4.2 million training triplets, it did not reach 80%.

**Controlling transformation dimensionality does not mitigate model misidentification.** One plausible cause for the limited model recovery is differences in the number of adjustable parameters: some models have more units in their final representational layer, resulting in a greater number of adjustable parameters in $\mathbf{W}$. These models might better fit the data regardless of its source. However, when we reran the simulations using only the top 500 principal components of each model's representation as features (thus fixing the parameter count in $\mathbf{W}$ to $500 \times 500$), model recovery accuracy did not improve (Fig. S3).

**Model recovery performance plateaus despite objective-driven representational divergence** Recent work by Lampinen et al. [49] showed that different training objectives induce distinct representational biases. To test how these biases affect model recovery, we expanded the model set to include 10 additional models, primarily image–text-aligned (Table S5). We then evaluated model recovery accuracy using the expanded model set (see Fig. S4). As expected, model recovery

accuracy declined with the expanded model set. It plateaued near 70%, even with 4.2 million training triplets. Next, we categorized each model as supervised, unsupervised, or image–text-aligned and performed a between-objective model recovery analysis (see Fig. S5). Even with 4.2 million training triplets, objective-based recovery reached only 73.7% (Fig. S5D), despite being significantly easier than model-based recovery. These results indicate that, although initial internal representations differ in objective-specific biases [49], linear probing can obscure objective-specific differences in representational geometry. Grouping models by architecture type (convolutional vs. vision transformers) yielded similar results (70.3% accuracy, Fig. S5).

**Certain models dominate recovery—even when incorrect.** Inspection of confusion matrices (Fig. 1B–D) indicates that the error is systematic: one model, OpenAI CLIP ResNet-50, was consistently misattributed as the ground-truth model. Would model recovery reach 100% accuracy if this model were excluded? To test this, we measured the mean rank of each model's predictive accuracy when it served as the data-generating model (Fig. 3A). Four of the 20 models had a mean rank above 2 (i.e., worse than second place), indicating that, on average, when these models generated the data, more than one competitor achieved higher predictive accuracy. We also computed the mean rank of each model's predictive accuracy when it was *not* the data-generating model (Fig. 3B). This revealed considerable variation in the models' propensity to be falsely identified as the data-generating model.

### 3.3 Representational geometry-based causes of model misidentification

The limited model-recovery accuracy prompted us to examine factors that might cause or modulate model misidentification. Specifically, we assessed how the representational geometry of each model, defined by the set of pairwise distances among its stimulus representations [36], was altered by linear probing. For each model, we computed a representational dissimilarity matrix (RDM) consisting of squared Euclidean distances among its final representational layer activation patterns in response to the 1,854 THINGS object images. This measurement was conducted both before and after aligning these representations to THINGS odd-one-out with a full $\mathbf{W}$. As a surrogate of human representations, we also included the RDM of VICE [41], an embedding model directly fitted to THINGS odd-one-out. We quantified within- and between-model RDM similarity using whitened Pearson correlation [50, 51] (Fig. S6) and employed multidimensional scaling (MDS) to summarize and visualize

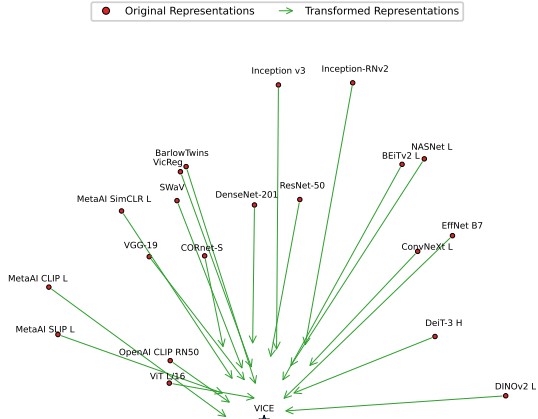

Figure 4: Shifts in model representations after linear probing, visualized using multidimensional scaling (MDS). Dots mark each model's original final representations. Arrowheads mark the aligned representations. VICE [41], an embedding model fitted to THINGS odd-one-out, serves as a proxy for human-like representation in this visualization.

these results (Fig. 4). As expected, all models' representational geometries shifted toward that of VICE following the alignment. Models that best predicted human judgments (e.g., ViT-L/16 and OpenAI CLIP-ResNet50) exhibited representational geometries more similar to VICE from the outset. By contrast, models that often failed to be recovered (e.g., EfficientNet B7, NASNet Large, or Inception-ResNet-V2) had initial representational geometries more distant from those of VICE and underwent substantial shifts in representational geometry as a result of the linear transformation.

**Substantial alignment-induced representational shifts are related to poor model recovery outcomes** To test whether alignment-induced representational shifts predict model-specific recovery outcomes, we used a linear regression analysis with shift magnitude and other geometric and architectural features (Table S2) as predictors. The dependent variable was defined as the difference in predictive accuracy between the data-generating model and one alternative candidate model, computed separately within each simulation. This accuracy difference served as a continuous measure of the separability of each model pair.

The analysis revealed three significant predictors (Bonferroni-corrected over 10,000 bootstrap tests): First, the shift magnitude of the candidate model—how much its geometry changed under alignment—positively predicted accuracy differences ($\beta = 0.495$, p-value $= 0.02$), suggesting that models more altered by linear probing were less predictive of responses generated by other models. Conversely, the shift magnitude of the data-generating model negatively predicted accuracy differences ($\beta = -0.2510$, p-value $= 0.01$), indicating that substantially adapted models yielded synthetic responses more easily predicted by other models than by themselves.

The third significant predictor was the *effective dimensionality* (ED) of the data-generating model representations after alignment to THINGS odd-one-out. ED quantifies how many feature space dimensions account for meaningful variance [52]; we used a standard estimator as detailed in Appendix A.5. Recent work has linked higher ED of neural network representations to improved prediction of visual cortical responses [53], though see [14, 54] for contrasting views. Higher post-transformation ED in the data-generating model negatively predicted accuracy differences ($\beta = -0.455$, p-value $= 0.01$), suggesting that models whose aligned representations are high-dimensional are less likely to be correctly recovered.

### 3.4 The predictive-accuracy–model-identifiability trade-off

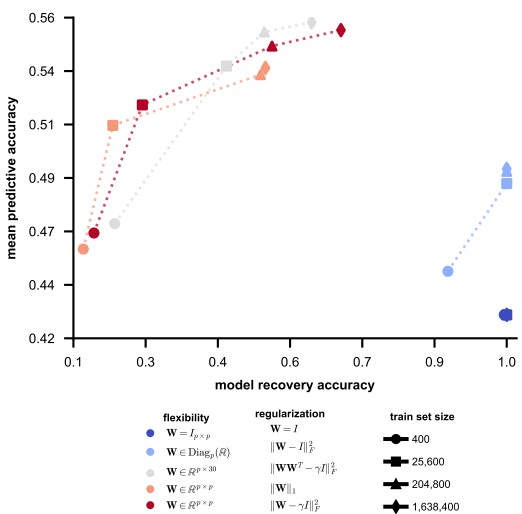

Figure 5: Model recovery accuracy vs. predictive accuracy (averaged across models), as a function of flexibility level and dataset size.

A seemingly straightforward solution to the problem of model misidentification is to restrict evaluation flexibility—either by using zero-shot predictions or by applying linear probing with fewer free parameters in the transformation matrix. However, the mean predictive accuracy across the twenty models drops markedly when they cannot reweight or linearly remix their features (Fig. 2A).

To characterize the trade-off between predictive accuracy and model identifiability, we repeated the model recovery experiments while varying the level of evaluation flexibility: from zero-shot evaluation, through diagonal and thin rectangular $\mathbf{W}$ matrices, to unrestricted linear probing. We matched flexibility constraints across the data-generation and evaluation stages: for example, if the data-generating model was fitted using a diagonal $\mathbf{W}$, candidate models (including the generating model) were also evaluated using diagonal matrices. This was done across multiple simulated dataset sizes. Similarly, we re-estimated empirical predictive accuracy using randomly subsampled subsets of THINGS odd-one-out.

As shown in Figure 5, there is a clear trade-off between predictive accuracy and model recovery accuracy. As evaluation flexibility increases, we gain predictive accuracy at the expense of discriminability.

## 4 Discussion

Our results show that, even with millions of trials, linear probing can fail to identify the model that generated the data. For our set of candidate models, model recovery accuracy plateaus below 80%. Holding the number of features constant across models—and thus the parameter count of the linear transformation matrix—does not mitigate the problem. Furthermore, in typical small-scale experiments (e.g., 100,000 trials), model recovery accuracy can be far worse—for our model set, it remains below 50%. These findings call into question the use of predictive accuracy under linear probing as an alignment metric for comparing models of biological representation.

**Limitations.**   The scope of our simulations is limited to behavioral data, and specifically, to the THINGS odd-one-out task. We chose this task as a test case because it is supported by a large empirical dataset [40] and allows straightforward simulation of synthetic responses by sampling from model-specified multinomial distributions. The noise-calibrated simulation approach can be readily extended to other behavioral paradigms, such as classification [15] or multi-arrangement (Kriegeskorte & Mur, 2012). Model identification using neural data operates in a markedly different regime: responses are multivariate and continuous, rather than univariate and discrete as in the behavioral case. Therefore, while our results demonstrate a pronounced predictivity–identifiability trade-off when comparing models to behavior in a large dataset, the severity of this trade-off for neural data cannot be inferred from our findings. Recent reports of qualitatively distinct neural network models achieving indistinguishable performance under flexible comparisons to neural data [10, 14] make this question especially pertinent. Addressing it will require future work using noise-calibrated, modality-specific neural simulations. It is important to note that the quantitative model recovery accuracy levels reported are specific to the candidate model set used. Still, we expect the qualitative finding of prevalent model misidentification to generalize to other model sets of similar size and to become more pronounced with larger sets.

A more fundamental limitation is that, in real-world comparisons between model predictions and empirical responses, the true model—the biological representations—is absent from the candidate set. Thus, model recovery within a closed set is a necessary but insufficient criterion for reliable model-comparison experiments.

**Navigating the accuracy–identifiability trade-off.**   The empirical findings highlight a tension between predictive performance and model identifiability: increasing the flexibility of the alignment metric improves predictive accuracy, but it also reduces the ability to discriminate among competing models. Experimental and analytical decisions guided solely by the goal of maximizing predictive accuracy risk overlooking this trade-off, thereby landing at its far end, where predictive performance is high but mechanistic correspondence to the modeled system is limited. Therefore, the pursuit of predictive performance must be tempered by attention to the specificity of the predictions: for example, through noise-calibrated model recovery simulations, as explored here.

Progress beyond the limitations of the accuracy–identifiability trade-off may require rethinking evaluation practices along three key directions.

*1. Change the stimuli:* As in many model comparison studies, we evaluated models out-of-sample— that is, on new stimuli drawn from the training distribution. Out-of-distribution generalization, which more strongly probes the models' inductive biases, may offer greater model comparative power [55, 56]. Stimuli designed to elicit model disagreement may yield even greater gains  [57–61].

The recovery gains we obtained from larger and more diagnostic triplet sets suggest that smarter sampling matters at least as much as sheer volume (Fig. 1). Adaptive, model-driven stimulus selection—constructing trials that maximize the expected divergence in network responses—can sharpen our ability to treat predictive accuracy as an indicator for human-model alignment, enhancing current flexible alignment methods without compromising identifiability [56–60].

*2. Change the metrics:* Constraining data-driven model alignment by biologically motivated and/or inter-individual variability–informed priors [23, 62–65] may improve upon the overly flexible family of linear transformations. Furthermore, imposing greater constraints on the readout may enhance its interpretability. For example, constraining the learned stimulus embeddings to be non-negative prevents features from canceling each other (e.g., [66]). Finally, Bayesian readout models, which estimate a distribution of feature weights rather than a point estimate, may improve robustness to sampling noise.

*3. Change the models:* The considerable geometric shifts required to align the networks suggest that linear probes can obscure important representational mismatches. Embedding richer priors directly into the models—through task design, objective functions, or biologically inspired architectures [5, 67– 71]—could allow aligned representations to emerge natively, reducing the need for substantial post hoc transformations. More broadly, further progress in neural network-based modeling of brain and behavior may depend less on ever-larger data-driven fits of pre-trained models and more on deliberate model refinement to embody explicit computational hypotheses.

---

**Code and data** are available on `github.com/brainsandmachines/oddoneout_model_recovery`

## Acknowledgments

This work was supported by the Israel Science Foundation (grant number 534/24 to T.G.).

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

# A Appendices

## A.1 Scalar-matrix shrinkage regularizer

**Lemma 1.** *Let* $\mathbf{W} \in \mathbb{R}^{p \times p}$. *Define*

$$\mathcal{R}(\mathbf{W}) = \min_{\gamma \in \mathbb{R}} \|\mathbf{W} - \gamma \mathbf{I}\|_F^2.$$

*Then*

$$\mathcal{R}(\mathbf{W}) = \|\mathbf{W}\|_F^2 - \frac{\mathrm{tr}(\mathbf{W})^2}{p}.$$

*Proof.* Using $\|\mathbf{A}\|_F^2 = \mathrm{tr}(\mathbf{A}^\mathsf{T}\mathbf{A})$,

$$\|\mathbf{W} - \gamma \mathbf{I}\|_F^2 = \mathrm{tr}[(\mathbf{W} - \gamma \mathbf{I})^\mathsf{T}(\mathbf{W} - \gamma \mathbf{I})] = \|\mathbf{W}\|_F^2 - 2\gamma\,\mathrm{tr}(\mathbf{W}) + \gamma^2 p = f(\gamma).$$

Because

$$f(\gamma) = p\gamma^2 - 2\,\mathrm{tr}(\mathbf{W})\,\gamma + \|\mathbf{W}\|_F^2, \qquad f'(\gamma) = 2p\gamma - 2\,\mathrm{tr}(\mathbf{W}), \qquad f''(\gamma) = 2p > 0,$$

the unique minimizer is

$$\gamma^\star = \frac{\mathrm{tr}(\mathbf{W})}{p}.$$

Substituting $\gamma^\star$ into $f(\gamma)$ yields

$$\min_\gamma f(\gamma) = \|\mathbf{W}\|_F^2 - \frac{\mathrm{tr}(\mathbf{W})^2}{p},$$

$\square$

**Notation.** Throughout, let $p \in \mathbb{N}$ be the dimension of the square matrix $\mathbf{W} \in \mathbb{R}^{p \times p}$; that is, $\mathbf{W}$ has $p$ rows and $p$ columns. Because $p$ counts rows/columns it satisfies $p \geq 1$, hence $p > 0$. This fact ensures the quadratic $f(\gamma) = p\gamma^2 - 2\,\mathrm{tr}(\mathbf{W})\gamma + \|\mathbf{W}\|_F^2$ is *strictly* convex: its second derivative is $f''(\gamma) = 2p > 0$, guaranteeing a unique minimizer $\gamma^\star$ in Lemma 1.

## A.2 Calibration to human noise ceiling

To ensure that the simulated responses were realistically distributed, we optimized each data-generating model's softmax temperature so that its simulated noise ceiling matches the empirical noise ceiling, estimated from a subset of the THINGS-odd-one-out dataset that includes responses to 1,000 triplets, each presented to approximately 30 participants.

### A.2.1 Noise ceiling estimation

Let $c_a^{(t)}$ denote the number of participants who chose the stimulus in position $a \in \{1, 2, 3\}$ as the odd-one-out for triplet $t \in \{1, \ldots, N\}$.

$$\mathbf{c}^{(t)} = (c_1^{(t)}, c_2^{(t)}, c_3^{(t)}), \qquad T^{(t)} = \sum_{i=1}^3 c_i^{(t)}.$$

For each $i \in \{1, 2, 3\}$, define

$$\mathbf{c}_{-i}^{(t)} = \mathbf{c}^{(t)} - \mathbf{e}_i, \quad V_i^{(t)} = \left\{ j \in \{1, 2, 3\} \mid c_{-i,j}^{(t)} = \max_k c_{-i,k}^{(t)} \right\},$$

where $\mathbf{e}_i$ is the $i$-th standard basis vector in $\mathbb{R}^3$. The leave-one-subject-out (LOO) accuracy for triplet $t$ is

$$a^{(t)} = \frac{1}{T^{(t)}} \sum_{i=1}^3 c_i^{(t)} \frac{\mathbf{1}\left[i \in V_i^{(t)}\right]}{|V_i^{(t)}|},$$

and the overall LOO noise ceiling is

$$NC_{\mathrm{LOO}} = \frac{1}{N} \sum_{t=1}^{N} a^{(t)}.$$

### A.2.2 Temperature calibration

We calibrated the softmax temperature $T$ of each data-generating model so that the model's predicted noise ceiling matches the one estimated from human data.

**Empirical noise ceiling.** Let $\eta_{\mathrm{ceil}} = 0.678$ denote the noise ceiling estimated from the THINGS odd-one-out triplet judgments using a specific repeated set across participants $D_{\mathrm{cal}}$. This triplet set was used exclusively during the calibration phase.

**Model-estimated noise ceiling.** Let $D_{\mathrm{cal}}$ be our calibration set of triplets drawn from the same pool. For each triplet $\{a, b, c\}_i \in D_{\mathrm{cal}}$, the model assigns the following probability:

$$p\big(\text{odd-one-out} = x \mid \text{triplet}_i\big) = \frac{\exp(S_{y,z}/T)}{\exp(S_{a,b}/T) + \exp(S_{a,c}/T) + \exp(S_{b,c}/T)}, \tag{7}$$

where $\{y, z\} = \{a, b, c\} \setminus \{x\}$. We then define the model's noise ceiling as the average, over all calibration triplets, of the model's maximum (top-choice) probability:

$$\hat{\eta}_{\mathrm{ceil}}(T) = \frac{1}{|D_{\mathrm{cal}}|} \sum_{i=1}^{|D_{\mathrm{cal}}|} \max_{x \in \{a,b,c\}} p\big(\text{odd-one-out} = x \mid \text{triplet}_i\big). \tag{8}$$

**Optimal temperature.** We choose $T$ to minimize the squared deviation between the empirical and model-estimated noise ceilings:

$$T^* = \arg\min_T \big[\, \eta_{\mathrm{ceil}} - \hat{\eta}_{\mathrm{ceil}}(T) \big]^2. \tag{9}$$

This procedure guarantees that, on average, a noise ceiling estimated from simulated responses to the calibration triplet set would match the empirical human noise ceiling.

### A.3 Sampling random triplets

To test model recovery under general conditions, the simulation study used random triplets *not* included in the THINGS odd-out dataset. In each simulation, we randomly partitioned the 1,854 images into three equally-sized disjoint subsets. One subset served as the test-image pool, and the remaining two subsets were concatenated and then randomly split into training (80%) and validation (20%) image pools. Using these pools, we randomly sampled 50 to 5.25 million triplets (spanning a logarithmically spaced range of stimulus set sizes), such that 80%, 10%, and 10% of the triplets were drawn from the training, validation, and test pools, respectively. No triplet included images from more than one split or overlapped with THINGS odd-one-out. Candidate model predictive accuracy (see Eq. 10) was evaluated and averaged across the three cross-validation folds, each using a different test-image pool assignment. Thus, in each simulation, each image appeared in the test pool in exactly one of the three cross-validation folds.

### A.4 Model-recovery formulation

This section describes the model recovery simulations in detail.

**Notation:**

- Let $\mathbf{W} \in \mathbb{R}^{p \times p}$ denote a linear transformation matrix that maps neural network representations into a target representational space (either behavioral or model-generated). This transformation is used to align models to human judgments or to other models' simulated responses.

- Let $M_i$ denote the $i$-th neural network model.

- Let $\mathcal{M} = \{M_1, \ldots, M_N\}$ be the set of $N$ pretrained encoders.

- Let $\tau$ denote the behavioral dataset consisting of human odd-one-out judgments.

- Let $\mathbf{W}_{M_i \rightarrow \star} \in \mathbb{R}^{p \times p}$ denote the linear transformation $\mathbf{W}$ which maps the $p$-dimensional features of $M_i$ into a target response space.

Throughout, the notation $\mathbf{W}_{\text{source} \rightarrow \text{target}}$ emphasizes that features from the model on the left are being aligned to judgments (or predictions) associated with the target on the right.

**Model-to-Behavior Alignment.** $\mathbf{W}_{M_i \rightarrow \tau}$ is the transformation matrix learned to map representations from model $M_i$ to best predict responses from the THINGS odd-one-out dataset. We use the shorthands $M_{i \rightarrow \tau}$ and $\mathcal{G}$:

$$M_{i \rightarrow \tau} := f(M_i, \mathbf{W}_{M_i \rightarrow \tau}), \qquad \mathcal{G} := \{M_{1 \rightarrow \tau}, \ldots, M_{N \rightarrow \tau}\}.$$

**Model-to-Model Alignment (Simulated Data).** Given a generator $M_{i \rightarrow \tau} \in \mathcal{G}$ and a candidate model $M_j \in \mathcal{M}$, we fit

$$\mathbf{W}^{(d)}_{M_j \rightarrow M_{i \rightarrow \tau}} \in \mathbb{R}^{p \times p}, \qquad M^{(d)}_{j \rightarrow i} := f(M_j, \mathbf{W}^{(d)}_{M_j \rightarrow M_{i \rightarrow \tau}}),$$

separately for every simulated dataset $d \in \{1, \ldots, D\}$ so that $M^{(d)}_{j \rightarrow i}$ best predicts the synthetic responses of $M_{i \rightarrow \tau}$.

**Model Recovery Experiments**

**Prediction vectors.** Let $d^{(k)} \in d$ be the $k$-th triplet in $d$. Let $y_{i \rightarrow \tau}(d^{(k)}) \in \{1, 2, 3\}$ be the true label of the $d^{(k)}$ triplet from generator $M_{i \rightarrow \tau}$. Similarly, let $\hat{y}_{j \rightarrow i}(d^{(k)})$ be the predicted labels of $M^{(d)}_{j \rightarrow i}$. For a full dataset $d$:

$$\hat{\mathbf{y}}_{j \rightarrow i}(d) = \{\hat{y}_{j \rightarrow i}(d^{(1)}), \ldots, \hat{y}_{j \rightarrow i}(d^{(K)})\},$$
$$\mathbf{y}_{i \rightarrow \tau}(d) = \{y_{i \rightarrow \tau}(d^{(1)}) \ldots, y_{i \rightarrow \tau}(d^{(K)})\},$$

where $K = |d|$.

**Candidate model predictive accuracy** For each $(M_i, M_j)$, let $\{d_{\text{train}}, d_{\text{test}}\}$ be a partition of dataset $d$. Candidate model predictive accuracy is defined by:

$$\text{Acc}(M^{(d_{\text{train}})}_{j \rightarrow i} | M_{i \rightarrow \tau}) = \frac{1}{|d_{\text{test}}|} \sum_{k=1}^{|d_{\text{test}}|} \left[ \hat{y}_{j \rightarrow i}(d^{(k)}_{\text{test}}) = y_{i \rightarrow \tau}(d^{(k)}_{\text{test}}) \right]. \tag{10}$$

where $\left[ \cdot \right]$ equals 1 if the condition is true and 0 otherwise.

**Model-recovery accuracy** The model recovery accuracy can be defined as:

$$\frac{1}{|\mathcal{M}|} \sum_{i=1}^{|\mathcal{M}|} \frac{1}{D} \sum_{d=1}^{D} \left[ \text{Acc}(M^{(d_{\text{train}})}_{i \rightarrow i} | M_{i \rightarrow \tau}) = \max_{j \in \{1, \ldots, |\mathcal{M}|\}} \text{Acc}(M^{(d_{\text{train}})}_{j \rightarrow i} | M_{i \rightarrow \tau}) \right] \tag{11}$$

This is the fraction of all simulations in which the candidate model with the highest predictive accuracy matches the true generator.

## A.5 Estimation of effective dimensionality

For each model, we compute the $D \times D$ covariance matrix of the activations in its deepest representational layer (where $D$ is the number of units in that layer, details in Appendix A.8), across the 1,854 images from the THINGS dataset, obtain the eigenvalues ($\lambda_i$) of this matrix using Principal Component Analysis (PCA), and then calculate:

$$ED = \frac{(\sum_{i=1}^{D} \lambda_i)^2}{\sum_{i=1}^{D} \lambda_i^2} \tag{12}$$

This quantity (labeled $n_2$ in [52]) estimates the participation ratio—the effective number of principal components contributing to the total variance [52, 53]. We repeated this procedure for model activations obtained after applying the human-aligned linear transformation (Section 2.2).

## A.6 Estimation of Intrinsic Dimensionality

We estimated intrinsic dimensionality (ID) with GRIDE (Generalized Ratios Intrinsic Dimension Estimator) [72], using its implementation in the `dadapy` Python package [73] with default settings. For each model, we extracted activations for the 1,854 THINGS images from its deepest representational layer (details in Appendix A.8), yielding a $1{,}854 \times D$ activation matrix (where $D$ is the number of units in that layer). We next applied the learned linear human-alignment transform $\mathbf{W}$—fit to predict the THINGS odd-one-out responses (Section 2.2)—and computed ID on both the original and transformed activations. Given $n$ data points $\{x_i\}_{i=1}^{n} \subset \mathbb{R}^D$, we compute Euclidean nearest-neighbor distances and, at multiple scales, form the ordered–neighbor distance ratios $\dot{\mu}_i \equiv \mu_{i,n_1,n_2} = r_{n_2}^{(i)}/r_{n_1}^{(i)}$ with $n_2 > n_1 \geq 1$ (the chosen neighbor orders that set the scale). GRIDE then estimates the intrinsic dimension $d$ by maximum likelihood under a local homogeneity (Poisson) assumption, maximizing the log-likelihood

$$\hat{d} = \arg\max \left[ n \log(d) + (n_2{-}n_1{-}1) \sum_i \log(\dot{\mu}_i^d{-}1) - \log(B(n_2{-}n_1,\, n_1)) - ((n_2{-}1)d{+}1) \sum_i \log(\dot{\mu}_i) \right]. \tag{13}$$

where $B(\cdot, \cdot)$ denotes the Beta function.

The final ID estimate, $\hat{d}$, is determined by identifying a stable "plateau" in the ID values across the different scales. This stable value represents the representation's intrinsic dimensionality, distinguishing it from noise that typically appears at very small scales.

## A.7 Resources

**Simulations.** We conducted large-scale simulations in which each of 20 models acted both as a data generator and as a candidate model. For each data-generating model, we sampled 30 random datasets per experimental condition (triplet set size).

Our main analysis covered 18 triplet-set sizes, yielding $18 \times 600 = 10{,}800$ simulations for the results shown in Fig. 1. For every run we selected the optimal regularization coefficient from ten values (logarithmically spaced from $10^{-6}$ to $10^5$) via cross-validation.

Simulations were conducted on the BGU ISE-CS-DT cluster, mainly using a server with eight NVIDIA RTX 6000 Ada GPUs. All preprocessing and analysis were performed on a local workstation. Simulation runtime depended on transformation flexibility, dataset size, model count, and random seed initialization. The mean runtime was approximately 9.5 minutes per simulation, so reproducing the 10,800 simulations of Fig. 1 on eight RTX 6000 GPUs would require roughly nine to ten days.

## A.8 Implementation details

**Feature extraction.** Feature vectors were extracted using an in-house Python package that wraps `torchvision`, Hugging Face, and TorchHub [74]. Meta AI models were obtained from the SLIP repository [75].

For all models, the deepest representational layer was extracted. Specifically, if the ultimate layer encoded predictions (e.g., logits in classifiers), we extracted the penultimate layer activations. If the

ultimate layer encoded embeddings (e.g., as in self-supervised or image-text-aligned models), we extracted the ultimate layer activations.

**RSA and MDS computations.** Representational similarity analyses (RSA) [20] and multidimensional scaling (MDS) were performed with the RSA Toolbox for Python [51].

## B Supplemental Figures

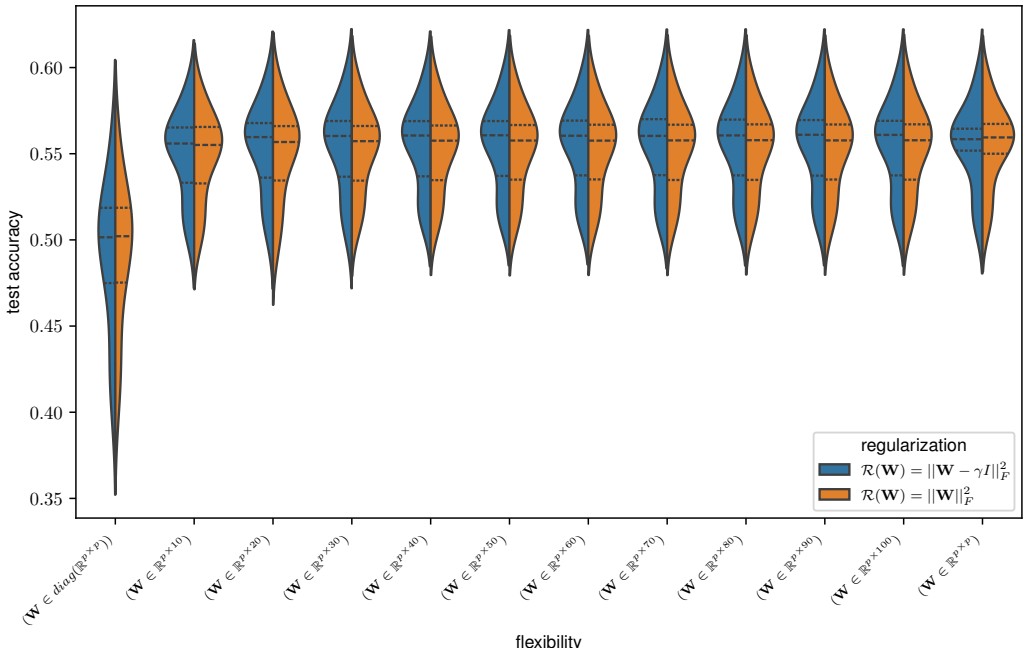

Figure S1: To verify that scalar-matrix shrinkage regularization (Eq. 4) does not impair model predictive accuracy, we evaluated all models on the THINGS odd-one-out dataset and compared the results obtained using Frobenius norm-based regularization to those obtained using scalar-matrix shrinkage regularization. Each violin plot depicts the distribution of odd-one-out predictive accuracy across models under the two regularization methods. The x-axis indicates levels of transformation matrix flexibility, from diagonal (left) to unconstrained (right). As evident from the overlapping distributions, we observed no meaningful differences in predictive accuracy between the two methods at any regularization level. A two one-sided tests (TOST) procedure with $\Delta = 0.05$ was used to assess the equivalence of the regularization methods' means; a Bonferroni correction for multiple comparisons was applied, indicating statistical equivalence across all levels of flexibility. Note, however, that this equivalence holds for optimal regularization. When the transformation is over-regularized, the scalar-matrix shrinkage regularization pulls the predictions toward the zero-shot solution, whereas the Frobenius norm-based regularization shrinks the transformation matrix toward the zero matrix, pulling the predictions toward a uniform distribution and thus impairing performance.

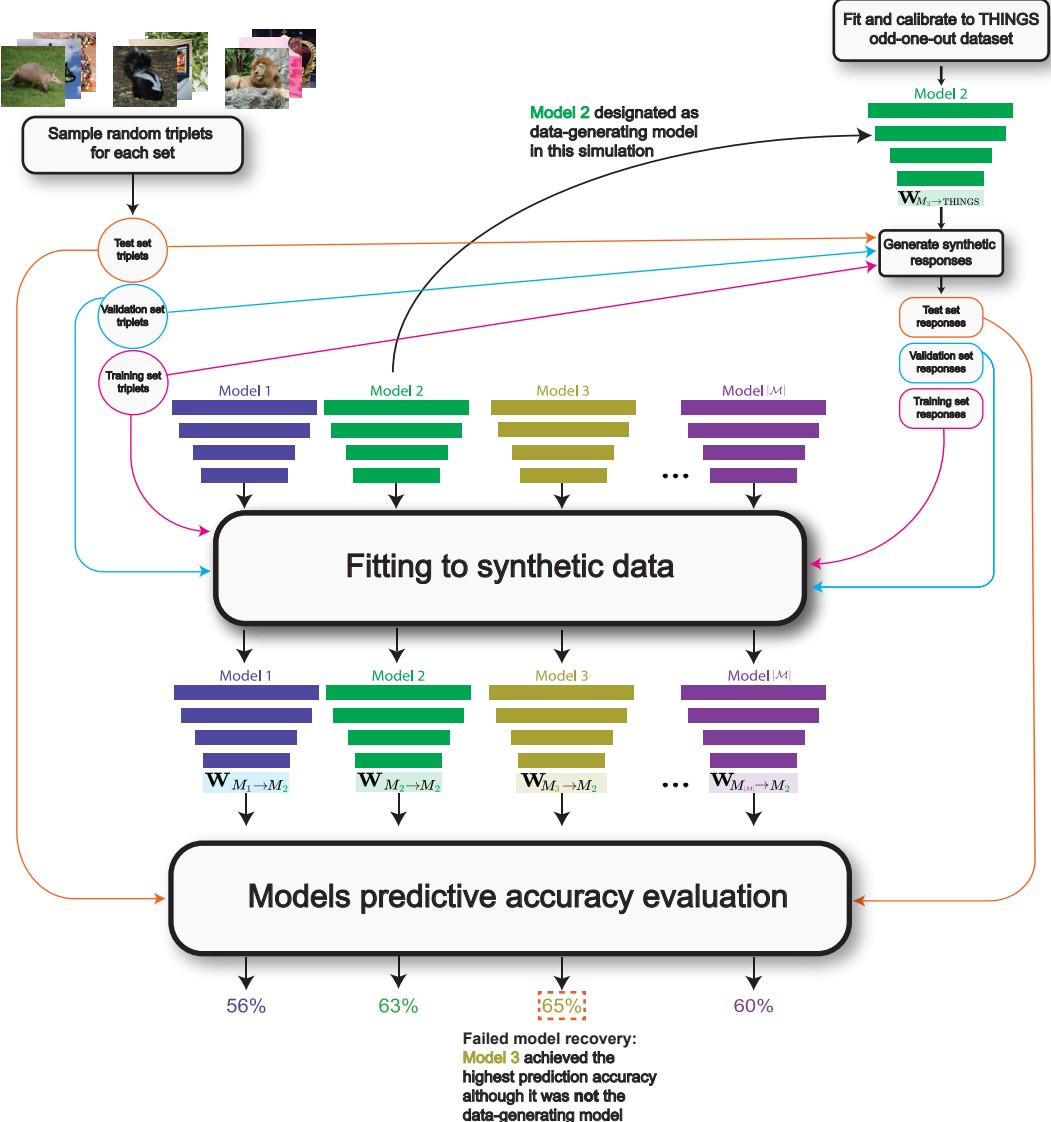

Figure S2: An illustration of one model recovery simulation. The designated data-generating model is fitted and calibrated to the THINGS odd-one-out dataset. Then, using three disjoint sets of images, we generate training, validation, and test sets of randomly sampled triplets with corresponding simulated responses. All candidate models, including the one that originally generated the data, are fitted with a model-specific linear transformation matrix using a regularization hyperparameter selected based on the validation set. After fitting, each model predicts the responses to the test set triplets, and its predictive accuracy is evaluated. This illustration demonstrates a case of misidentification, where the model that most accurately predicts the synthetic data is not the model that originally generated it. The natural images used in this illustration were taken from a CC0-licensed set of 1,854 images corresponding to the same concepts as in THINGS [40], included in THINGS+ [76].

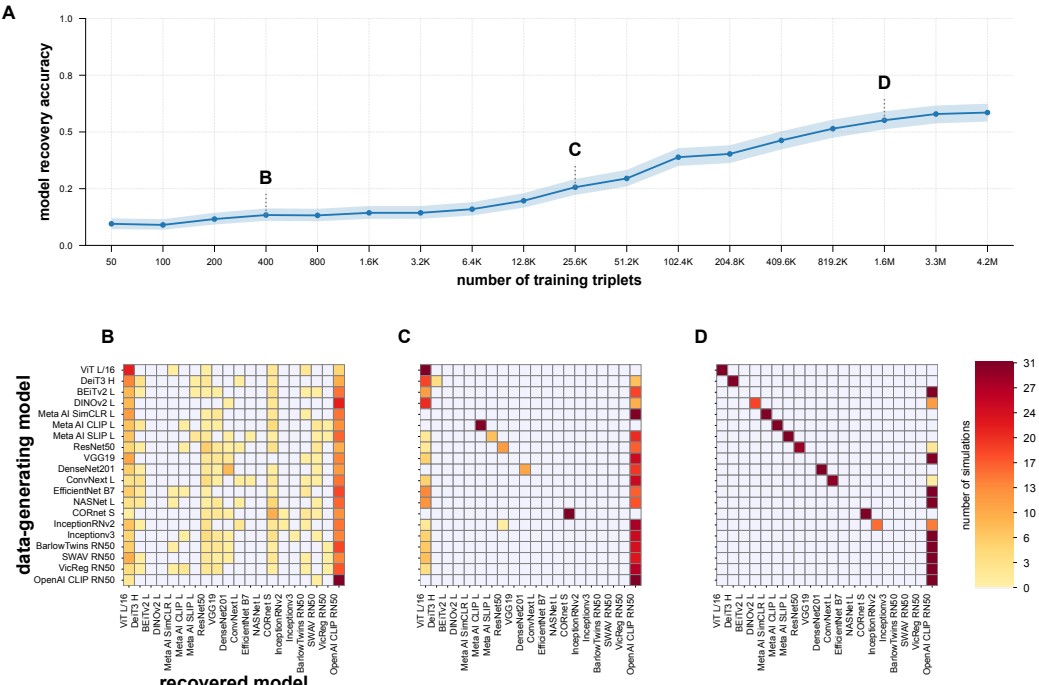

Figure S3: Same model recovery analysis as in Figure 1, while using the top 500 principal components (PCs) of each model's representation instead of its original features. For each model, we conducted principal component analysis on its final representational layer activation patterns in response to the ImageNet-1K validation set images and retained only the scores of the top 500 PCs. This dimensionality reduction was applied to the models' representations both when the models generated the data and when served as candidate models. Consequently, all fits used $500 \times 500$ transformation matrices.

(**A**) Model recovery accuracy under linear probing with a $500 \times 500$ matrix across different dataset sizes. (**B, C, D**) Confusion matrices at three training set sizes (400, 25.6K, and 1.6M triplets). Each matrix row corresponds to a data-generating model, and each matrix column to a recovered model. Diagonal entries represent correct model recovery. **Between-model differences in transformation dimensionality do not explain model misidentification.**

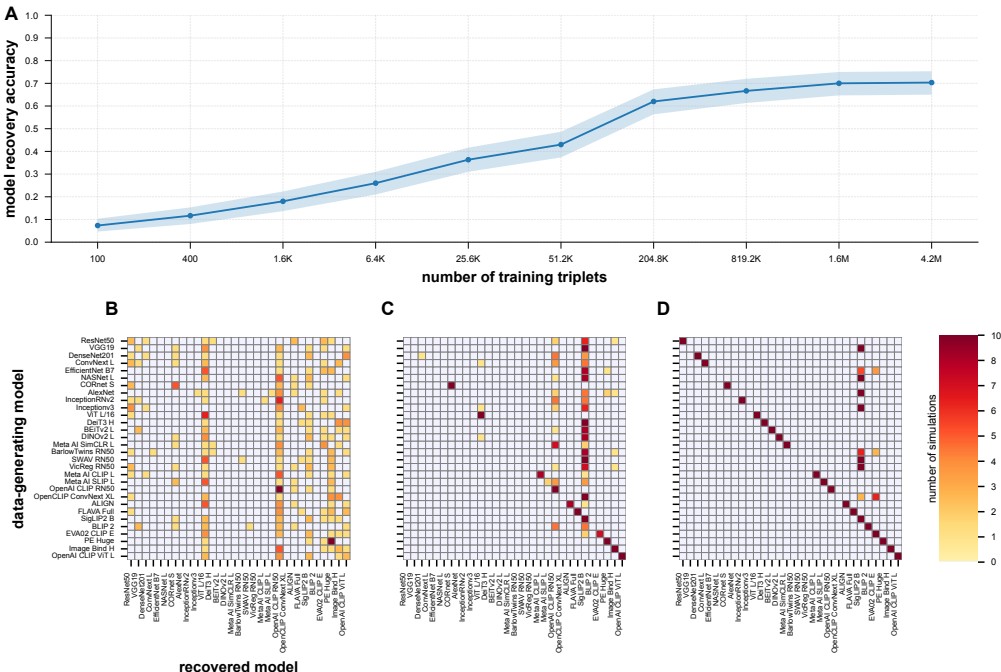

Figure S4: **Model recovery simulations with an extended model set.** We reran the model recovery simulations with 10 additional models (see Table S5), most of which were image–text aligned. To limit the computational cost, we used 10 simulations per experimental condition compared to 30 used in the main analysis (Fig. 1), and 10 different training dataset sizes compared to 18. Other than these changes, the analysis is the same as in Figure 1. (**A**) Model recovery accuracy under linear probing across different training set sizes. (**B–D**) Confusion matrices at three training set sizes (400, 25,600, and 1.6 million). As expected, adding more models reduced model recovery accuracy: it plateaued around 70%, compared to 80% in the original 20-model experiment (Fig. 1).

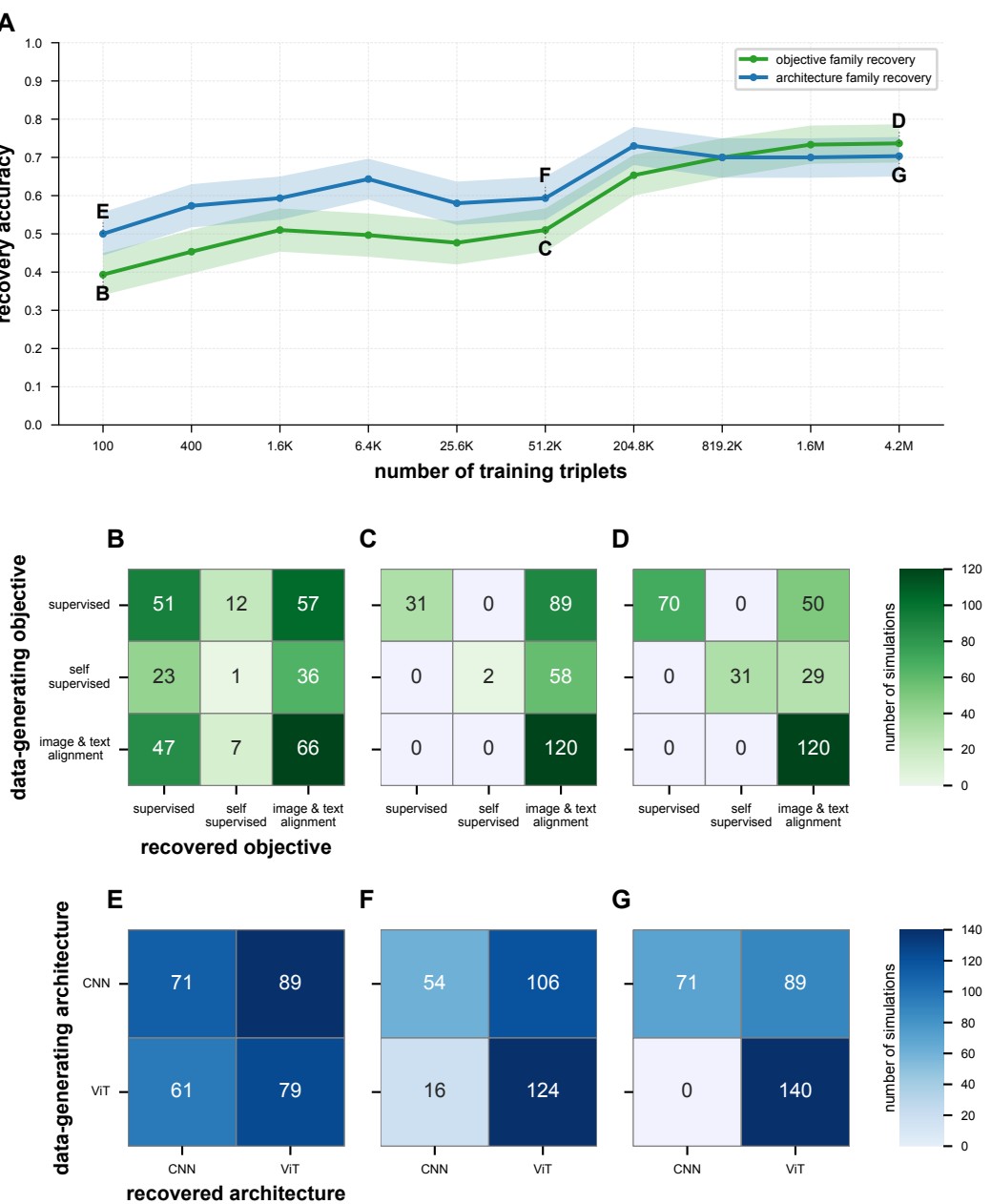

Figure S5: **Model recovery accuracy analysis, grouped by objective or architecture type.** Using the extended-model set simulations described in Figure S4, we inspected how well coarser model characteristics—objective type or architecture type—can be recovered. For objective type, we categorized each model's objective as supervised, self-supervised, or image–text alignment. For architecture type, we categorized each model's architecture as convolutional or vision transformer. We then evaluated model recovery accuracy at the category level, defining correct recovery as cases where the best-performing candidate model belonged to the same group as the data-generating model. Panel **A** shows model recovery accuracy across different training set sizes. The *green* line indicates recovery accuracy between objective types, and the *blue* line indicates recovery accuracy between architecture types. The grouped recovery accuracy for only 100 training triplets (confusion matrices shown in **B** and **E**) exceeded that of the non-grouped analysis (Fig. S4), owing to the higher chance level associated with those conditions. Even with 4.2 million training triplets (confusion matrices shown in **D** and **G**), the accuracy reached only about 72% for objective and 70% for architecture type. This result indicates that linear probing may obscure not only individual model identity but also broader representational motifs.

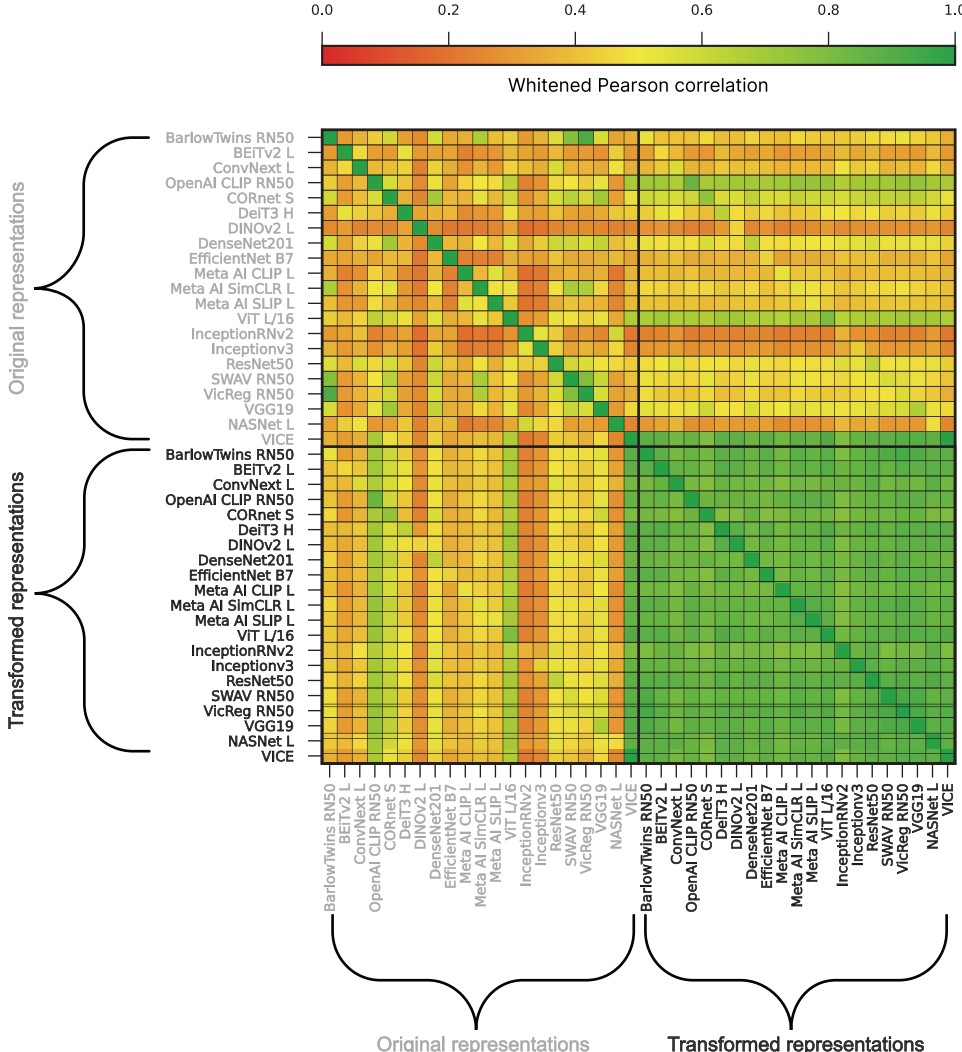

Figure S6: **Similarity of model representations before and after alignment to THINGS odd-one-out.** For each network, we computed a squared Euclidean representational dissimilarity matrix (RDM; $1{,}854 \times 1{,}854$) on the THINGS images and vectorized its upper-triangular entries. Each cell above shows the whitened Pearson correlation between two such RDM vectors—that is, the ordinary linear correlation after whitening by the sample covariance of RDM entries, which discounts shared-variance artifacts [50, 51]. Rows and columns are arranged in pairs: the first entry represents the model's *original* features, and the second entry its *transformed* features obtained from the alignment to THINGS odd-one-out, estimated as part of our recovery experiments. The embeddings of VICE [41] were used without any further fitting, since this model was trained to fit human odd-one-out responses. **Between-model dissimilarities:** The upper-left quadrant shows that unaligned models occupy distinct representational geometries; the off-diagonal correlations span a broad range (typically 0.2–0.5), reflecting diverse unaligned model representations. The lower-right quadrant displays a bright band of high off-diagonal similarity, indicating that alignment drives disparate models toward a shared geometry, eroding their "individuality." **Within-model representational shift:** The block-diagonal formed by each model's original vs. transformed RDMs reveals how far a model's geometry moves under the linear transform. We quantify this shift as $d_{\text{shift}} = 1 - \rho_{\text{whitened}}\big(\text{RDM}_{\text{orig}}, \text{RDM}_{\text{aligned}}\big)$, so larger values reflect greater internal reorganization. These within-model shift scores serve as the "alignment-induced representational shift" predictor in the regression analysis (see Table S2) and are visualized as green arrows in Figure 4.

# C  Supplemental tables

| Model | Objective | Zero-shot $\mathbf{W} = \mathbf{I}_{p \times p}$ | Diagonal $\mathbf{W} \in \mathrm{Diag}(\mathbb{R}^{p \times p})$ | Rectangular$_{30}$ $\mathbf{W} \in \mathbb{R}^{p \times 30}$ | Full $\mathbf{W} \in \mathbb{R}^{p \times p}$ | Reference |
|---|---|---|---|---|---|---|
| OpenAI CLIP RN50 | Image/Text contrastive | 0.4784 | 0.5577 | 0.5941 | 0.5959 | [45] |
| ViT L/16 | Image classification | 0.5370 | 0.5558 | 0.5937 | 0.5927 | [77] |
| DeiT3 H | Image classification | 0.4651 | 0.5201 | 0.5809 | 0.5786 | [78] |
| Meta AI SLIP L | Image–Text contrastive | 0.4386 | 0.4865 | 0.5751 | 0.5761 | [75] |
| Meta AI SimCLR L | Self-supervised contrastive | 0.3883 | 0.5095 | 0.5690 | 0.5693 | [75] |
| DINOv2 L | Self-distillation | 0.4127 | 0.5282 | 0.5721 | 0.5645 | [79] |
| Meta AI CLIP L | Image–Text contrastive | 0.4406 | 0.4882 | 0.5628 | 0.5636 | [75] |
| SwAV RN50 | Self-supervised clustering | 0.4118 | 0.5015 | 0.5635 | 0.5616 | [80] |
| BEiTv2 L | Masked image modeling | 0.4176 | 0.5133 | 0.5634 | 0.5605 | [81] |
| VicReg RN50 | VIC regularization | 0.4387 | 0.5186 | 0.5610 | 0.5584 | [82] |
| VGG19 | Image classification | 0.4679 | 0.5214 | 0.5583 | 0.5567 | [83] |
| ResNet50 | Image classification | 0.4568 | 0.5080 | 0.5584 | 0.5562 | [84] |
| BarlowTwins RN50 | Redundancy-reduction SSL | 0.4360 | 0.5072 | 0.5603 | 0.5559 | [85] |
| CORnet-S | Image classification | 0.4308 | 0.4815 | 0.5530 | 0.5526 | [67] |
| DenseNet201 | Image classification | 0.4323 | 0.4961 | 0.5523 | 0.5518 | [86] |
| ConvNeXt L | Image classification | 0.4493 | 0.4749 | 0.5367 | 0.5347 | [87] |
| EfficientNet B7 | Image classification | 0.4088 | 0.4601 | 0.5281 | 0.5249 | [88] |
| Inceptionv3 | Image classification | 0.3817 | 0.4198 | 0.5206 | 0.5184 | [89] |
| NASNet L | Image classification | 0.3799 | 0.4344 | 0.5158 | 0.5110 | [90] |
| InceptionRNv2 | Image classification | 0.3689 | 0.3989 | 0.5124 | 0.5067 | [89] |

Table S1: Candidate models used in this study and their cross-validated prediction accuracy on the THINGS odd-one-out dataset under varying alignment flexibility: zero-shot ($\mathbf{W} = \mathbf{I}_{p \times p}$), diagonal, rank-30 rectangular, and full square transformation. Features were extracted from each model's final representational layer (dimension $p$) and evaluated via 3-fold cross-validation over disjoint image sets. To ensure each model was evaluated under favorable conditions, we used its largest publicly available variant. The model set spans self-supervised, image-classification, and image–text alignment objectives. Note that models with substantial architectural and functional differences can achieve similar predictive performance.

| Model Name | #Parameters | #Features | Original features ED | Transformed features ED | Alignment-induced representational shift | Original features ID | Transformed features ID |
|---|---|---|---|---|---|---|---|
| BarlowTwins RN50 | 25557032 | 2048 | 240.84 | 136.77 | 0.49 | 29.7 | 36.98 |
| BeitV2 Large | 303405568 | 1024 | 224.07 | 57.19 | 0.53 | 23.13 | 20.51 |
| ConvNeXt Large | 197767336 | 1536 | 103.15 | 48.25 | 0.45 | 27.64 | 22.53 |
| OpenAI CLIP RN50 | 102007137 | 1024 | 47.20 | 16.13 | 0.17 | 17.37 | 14.49 |
| CORnet-S | 53416616 | 512 | 73.06 | 23.93 | 0.26 | 20.62 | 18.25 |
| Deit3 Huge | 630845440 | 1280 | 191.43 | 59.81 | 0.38 | 16.13 | 16.94 |
| DINOv2 Large | 304368640 | 1024 | 476.25 | 74.21 | 0.54 | 26.94 | 25.35 |
| DenseNet201 | 20013928 | 1920 | 154.78 | 48.20 | 0.37 | 28.006 | 24.63 |
| EfficientNet B7 | 66347960 | 2560 | 138.40 | 218.38 | 0.55 | 23.43 | 36.44 |
| MetaAI CLIP Large | 367254017 | 512 | 87.31 | 20.88 | 0.49 | 27.14 | 20.1 |
| MetaAI SimCLR Large | 325346560 | 1024 | 52.40 | 38.06 | 0.48 | 18.9 | 23.71 |
| MetaAI SLIP Large | 389298945 | 512 | 101.65 | 35.37 | 0.48 | 25.69 | 22.98 |
| ViT L/16 | 303301632 | 1024 | 88.30 | 50.01 | 0.21 | 22.49 | 22.69 |
| InceptionRNV2 | 54306464 | 1536 | 68.36 | 53.23 | 0.62 | 20.11 | 19.61 |
| InceptionV3 | 27161264 | 2048 | 174.87 | 77.88 | 0.56 | 30.46 | 25.55 |
| ResNet50 | 25557032 | 2048 | 130.50 | 84.88 | 0.39 | 28.27 | 24.73 |
| SWAV RN50 | 25557032 | 2048 | 140.06 | 133.86 | 0.43 | 26.17 | 37.58 |
| VicReg RN50 | 23508032 | 2048 | 225.99 | 119.89 | 0.45 | 29.87 | 33.52 |
| VGG19 | 143667240 | 4096 | 160.72 | 134.09 | 0.34 | 25.43 | 29.41 |
| NASNet Large | 84720150 | 4032 | 258.65 | 117.97 | 0.52 | 33.40 | 27.57 |

Table S2: Model characteristics used as predictors in the regression analyses. For each model, we recorded the following predictors:

**#Parameters –** total number of trainable parameters across all model layers.

**#Features –** number of units in the model's final representational layer.

**Original features ED –** the effective dimensionality (ED; see Appendix A.5) of the model's final representational layer activation patterns in response to the 1,854 THINGS images, measured before any transformation.

We included these first three predictors to test whether there are inherent, non-alignment-related properties of each model that play a role in model misidentification.

**Transformed features ED –** the effective dimensionality of the model's final representational layer, obtained after applying the fitted and calibrated linear transformation $\mathbf{W}$.

**Alignment-induced representational shift –** the representational dissimilarity between original and transformed feature spaces derived from the representational similarity analysis (RSA; Fig. S6), which captures the representational shift induced by the linear transformation.

We included the latter two predictors to test whether there are representational alignment-induced geometric changes that are associated with model misidentification.

As an alternative to effective dimensionality, we also considered the following two predictors:

**Original features ID –** the intrinsic dimensionality (ID; see Appendix A.6) of the model's final representational layer, obtained before any transformation.

**Transformed features ID –** the intrinsic dimensionality of the model's final representational layer, obtained after applying the fitted and calibrated linear transformation $\mathbf{W}$.

To avoid multicollinearity, the effective dimensionality measures were included in the analysis reported in Table S3, and the intrinsic dimensionality predictors were included in a separate regression analysis, reported in Table S4.

All model characteristics were extracted or estimated directly from the specific model implementations we used to ensure accuracy and reproducibility. Prior to conducting the regression analyses, the feature tables were expanded to account for all 380 pairwise model comparisons in each of the 30 simulations, yielding 11,400 observations and 10 predictors.

| Predictor | Candidate/Generator | $\beta$ | 95 % CI | p-value |
|---|---|---|---|---|
| **Transformed features ED** | Data generating model | $-0.455$ | [–0.840, –0.175] | 0.02 |
| **Transformed features ED** | Candidate model | $-0.182$ | [-0.967, 0.079] | 1 |
| **Original features ED** | Data generating model | $0.069$ | [–0.259, 0.181] | 1 |
| **Original features ED** | Candidate model | $-0.008$ | [-0.371, 0.328] | 1 |
| **#Features** | Data generating model | $-0.208$ | [–0.419, 0.059] | 0.89 |
| **#Features** | Candidate model | $0.264$ | [ -0.072, 0.764] | 0.924 |
| **Alignment-induced representational shift** | Data-generating | $-0.228$ | [–0.442, –0.117] | 0.02 |
| **Alignment-induced representational shift** | Candidate model | $0.495$ | [ 0.286, 0.841] | 0.03 |
| **#Parameters** | Data generating model | $0.035$ | [–0.122, 0.165] | 1 |
| **#Parameters** | Candidate model | $-0.145$ | [–0.454, 0.077] | 1 |

Table S3: **Regression analysis identifying drivers of misidentification.**
To determine which model properties (Table S2) may cause a model to be misidentified by its own simulated responses, we regressed the pairwise accuracy gap (retrained data-generating model − candidate model; negative values indicate misidentification) on model features across 11,400 comparisons from 600 simulations (20 models × 30 datasets × 20 candidates − 20 × 30 same model comparisons). We used the largest simulated dataset condition, each including 4.2 million training triplets. Predictors for both data-generators and candidates comprised:
**(1–2)** effective dimensionality (ED) on THINGS stimuli before and after linear alignment.
**(3)** The number of units in the model's final representational layer.
**(4)** Alignment-induced representational shifts between original and transformed feature spaces derived from the model similarity matrix (Fig. S6).
**(5)** Total number of trainable parameters.
As we used these predictors for every given pair of data-generating and candidate models in each simulation, we obtained a total of ten regression coefficients. An intercept term was included. To estimate uncertainty, we performed 10,000 bootstrap replicates at the model level: each replicate resampled with replacement, the 20 model identities from the original set to define both the data-generator and candidate pools (allowing duplicates), retained all 30 random seed initializations, and refit the regression to obtain empirical coefficient distributions. Table entries report the standardized coefficient ($\beta$), its 95% percentile bootstrap confidence interval, and bootstrap-based p-value, corrected for the ten coefficients. Candidate model dissimilarity ($\beta = 0.495$, p-value $= 0.02$) and data-generating model post-alignment ED ($\beta = -0.455$, p-value $= 0.02$) are the strongest significant drivers of misidentification, highlighting the critical role of transformation-induced geometric shifts.

| Predictor | Candidate/Generator | $\beta$ | 95 % CI | p-value |
|---|---|---|---|---|
| **Transformed features ID** | Data generating model | −0.401 | [–0.78, 0.28] | 0.2 |
| **Transformed features ID** | Candidate model | −0.37 | [-0.76,-0.09 ] | 0.17 |
| **Original features ID** | Data generating model | 0.08 | [–0.11, 0.34] | 1 |
| **Original features ID** | Candidate model | −0.01 | [-0.21, 0.28] | 1 |
| **#Features** | Data generating model | −0.39 | [–0.71, -0.20] | 0.03 |
| **#Features** | Candidate model | 0.26 | [ 0.04, 0.60] | 0.3 |
| **Alignment-induced representational shift** | Data-generating | −0.31 | [–0.76, –0.15] | 0.03 |
| **Alignment-induced representational shift** | Candidate model | 0.65 | [ 0.41,1.11 ] | 0.004 |
| **#Parameters** | Data generating model | 0.20 | [–1.02, 1.8] | 1 |
| **#Parameters** | Candidate model | −1.2 | [–2.82,0.17 ] | 0.78 |

Table S4: **Regression Analysis Using Intrinsic Dimensionality Measures.**
To further explore which model properties (Table S2) may cause a model to be misidentified by its own simulated responses, we ran the same regression analysis as in Table S3, replacing the ED measures with ID measures. None of the ID measures—for either the candidate or data-generating model, and for both the original and transformed features—were significant. Alignment-induced representational shift became significant for both the candidate model ($\beta = 0.65$, p-value = 0.004) and the data-generating model ($\beta = -0.31$, p-value = 0.03), highlighting the effect of the alignment-induced shift on model misidentification.

| Model | Objective | Zero-shot $\mathbf{W} = \mathbf{I}_{p \times p}$ | Diagonal $\mathbf{W} \in \text{Diag}(\mathbb{R}^{p \times p})$ | Rectangular$_{30}$ $\mathbf{W} \in \mathbb{R}^{p \times 30}$ | Full $\mathbf{W} \in \mathbb{R}^{p \times p}$ | Reference |
|---|---|---|---|---|---|---|
| OpenCLIP ConvNeXt XL | Image–Text contrastive | 0.4267 | 0.5065 | 0.5684 | 0.5599 | [91] |
| ALIGN | Image–Text contrastive | 0.4252 | 0.5274 | 0.5957 | 0.59324 | [92] |
| FLAVA Full | Image–Text contrastive | 0.4738 | 0.4738 | 0.5833 | 0.5834 | [93] |
| SigLIP2 B | Image–Text contrastive | 0.4497 | 0.5668 | 0.6114 | 0.6120 | [94] |
| AlexNet | Image classification | 0.4518 | 0.4900 | 0.5356 | 0.5300 | [95] |
| BLIP 2 | Image–Text contrastive | 0.3758 | 0.3758 | 0.4063 | 0.5192 | [96] |
| EVA02 CLIP Enormous | Image–Text contrastive | 0.5030 | 0.5686 | 0.6128 | 0.6108 | [97] |
| PE Huge | Image–Text Contrastive | 0.4803 | 0.5723 | 0.6129 | 0.6100 | [98] |
| Image Bind Huge | Multimodal Contrastive | 0.4621 | 0.4621 | 0.5336 | 0.6069 | [99] |
| OpenAI CLIP ViT | Image–Text Contrastive | 0.4191 | 0.5642 | 0.6057 | 0.60482 | [45] |

Table S5: Prediction accuracies of the 10 additional models used in the small-scale model recovery simulations for the grouped model recovery results presented in figure S5. The models were evaluated on the THINGS odd-one-out triplets using cross-validated prediction accuracies under varying alignment flexibilities: zero-shot ($W = I_p$), diagonal, rank-30 rectangular transform, and full square transform. Features from each model's final representational layer (dimension $p$) were evaluated using 3-fold cross-validation over disjoint image sets. To ensure each model was evaluated under favorable conditions, we used its largest publicly available variant.

