# OpenReview forum: "Model–Behavior Alignment under Flexible Evaluation: When the Best-Fitting Model Isn’t the Right One"
_NeurIPS.cc/2025/Conference — NeurIPS 2025 poster_

### Official Review · Reviewer_GT1J · 2025-06-30

**Clarity:** 4
**Significance:** 3
**Originality:** 4
**Rating:** 6
**Confidence:** 5

**Summary:**

The authors show that when pretrained neural network representations are fitted to predict human similarity judgements, these representations lose their identifiability and recoverability. While the situation is improved by an increase in dataset size, the ceiling caps at 80%.

Further, the authors conduct a set of interesting analyses trying to understand the factors that underlie the problem. They find that a high intrinsic dimensionality and a large change in representational geometry are linked to lower recoverability. They also show that one successful model (CLIP) is recovered from the synthetic data generated by several other models.

**Questions:**

- I see that you fit $W$ several times in model recovery across seeds, but it seems to be fitted just once in the actual model fitting. Could some of the unrecoverability be attributed to stochastic optimisation? How similar would the learned representations be if you fitted the same linear probe several times?
- Is $W$ initalised as an identity matrix in the model fitting as well?
- 80% recoverability is deemed not good in the paper. Intuitively, I agree with this. However, I wonder if you gave the same triplets to human participants several times, what the test-retest reliability would look like. I know it's not the same as recovery, and that there are other factors here, but I think it could be a good and interesting proxy. I know it's not in the scope of this paper, but it might be interesting work for the future.

**Ethical Concerns:**

["NO or VERY MINOR ethics concerns only"]

**Final Justification:**

I’m convinced the results are of great importance to the comp cog sci and neuro AI communities.

**Limitations:**

The authors have adequately discussed the limitations of their work.

**Paper Formatting Concerns:**

No formatting concerns.

**Quality:**

3

**Strengths And Weaknesses:**

# Strengths

I think this is an excellent paper. The problem addressed is of high relevance to the representational alignment community, and the results are very interesting (also concerning). While the main result is already exciting and accessible, I think the authors do a great job of digging into the causes behind the main findings. The writing is clear, and the fitting and evaluation methods are sound.

As an additional plus, the code provided by the authors is very well documented. I haven't had a chance to run it, but it definitely increases my belief in the accuracy of the reported results.

# Weaknesses

## Interpretation of Results in High-Data Regime

While both recoverability and identifiability are poor in low-data regimes, they are actually very good given a lot of data (ie, Figure 1D). The models that are not recovered well also seem to be outdated models (eg VGG19, EfficientNet, NasNet, Inception, BarlowTwins). If these models are not included, you would have almost a diagonal matrix in 1D. I think it's good you included these models, and I am not asking you to remove them. However, this needs to be carefully interpreted when thinking about how much of a practical problem unrecoverability is. e.g., if I compare a CLIP model, Imagenet ViT-L, and a DINO v2, the risk does not appear to be high. Nevertheless, I agree with the authors that such a high data regime is often not achieved in most studies. I just wanted to point this out for discussion.

## Measure of Effective Dimensionality

I know the PC based effective dimensionality metric is commonly used. However, I have my doubts about it. This paper [1] has shown that DNNs' representations lie on curved manifolds, which makes the PC based method inadequate due to its linear nature. I would recommend the authors use the Two-NN or the Gride method [2, 3], which does not make the linearity assumption. Either method can be easily used [with this package](https://github.com/sissa-data-science/DADApy), and would provide more accurate estimates.


---

Although I have outlined two weaknesses above, I think these are minor. The reason I'm giving a 5 and not a 6 is because I think the results are highly relevant to the representational alignment community, but I'm not sure if it has an impact in other areas. **I'd be willing to increase my score if the authors address the minor concerns above (the first one simply in writing, and the second one through simple analyses) and can make a convincing argument for why this work is relevant to a wider community.**



# References

[1] - Ansuini, A., Laio, A., Macke, J. H., & Zoccolan, D. (2019). Intrinsic dimension of data representations in deep neural networks. In H. Wallach, H. Larochelle, A. Beygelzimer, F. d\textquotesingle Alché-Buc, E. Fox, & R. Garnett (Eds.), Advances in Neural Information Processing Systems (Vol. 32).

[2] - Facco, E., d’Errico, M., Rodriguez, A., & Laio, A. (2017). Estimating the intrinsic dimension of datasets by a minimal neighborhood information. Scientific Reports, 7(1), 12140. doi:10.1038/s41598-017-11873-y

[3] - Denti, F., Doimo, D., Laio, A., & Mira, A. (2022). The generalized ratios intrinsic dimension estimator. Scientific Reports, 12(1), 20005. doi:10.1038/s41598-022-20991-1

---

> ### Author Rebuttal · Authors · 2025-07-31
>
> We are grateful for the reviewer's constructive and insightful review of our work.
>
> **1. Interpretation of Results in High-Data Regime.** We agree that in the high-data regime, misidentification affected smaller convolutional models, which tend to have lower capacity and were trained on smaller datasets, potentially yielding less distinctive representations. However, in a follow-up analysis including ten additional models (see our response to Reviewer Suzu), we found that even OpenCLIP-ConvNeXt-XL was often misidentified. This finding suggests that misidentification in the high-data regime may not be limited to older or lower-capacity models. Importantly, model confusability is context dependent: it is a property of model pairs rather than of individual models.
>
> This also raises a second concern. Suppose the read-out human representation is more similar to that of a simpler or smaller model. If flexible evaluation procedures tend to favor higher-capacity models because they can better approximate arbitrary targets, these procedures may systematically mislead us. In such cases, richer models could outscore simpler but more human-like ones—not because they are better aligned in their native geometry, but because they are more easily aligned post hoc. Therefore, we do not find the fact that it is mostly smaller CNNs being misidentified to be entirely reassuring.
>
> Due to space constraints in the current draft (even with the additional page), we cannot commit to a specific revision of the discussion section at this stage. However, we will incorporate this more nuanced interpretation, including the reviewer’s observation, into the revised version.
>
> **2. Measure of Effective Dimensionality.** We thank the reviewer for the helpful suggestion to explore non-linear dimensionality estimates, which we were glad to incorporate.
>
> Following this suggestion, we estimated intrinsic dimensionality (ID) using Generalized Ratios Intrinsic Dimensionality Estimator (GRIDE) (Denti et al., 2022) through the publicly available DADApy implementation (with GRIDE computed across neighborhood sizes k = 1–64).
>
> We computed ID for each model’s representation before and after alignment, and reran the regression analyses described in Table S3 in the manuscript, replacing PCA-based effective dimensionality with the GRIDE measure.
>
> Below are the updated results:
> | Predictor                                | Candidate/Generator                | β     | 95 % CI        | _p_-value |
> | ---------------------------------------- | --------------------- | ----- | -------------- | --------- |
> | # Parameters                             | Data‑generating model | 0.03  | [‑0.28, 0.28]  | 1.00      |
> | # Parameters                             | Candidate model       | ‑0.21 | [‑0.18, 0.3]   | 1.00      |
> | # Features                               | Data‑generating model | ‑0.37 | [‑0.44, 0.035] | 0.81      |
> | # Features                               | Candidate model       | 0.25  | [‑0.68, 0.2]   | 0.06      |
> | Original features ID                  | Data‑generating model | 0.09  | [‑0.14, 0.37]  | 1.00      |
> | Original features ID                  | Candidate model       | 0.01  | [-0.23, 0.31]  | 1.00      |
> | Transformed features ID               | Data‑generating model | -0.31 | [-0.61, ‑0.05] | 0.2       |
> | Transformed features ID               | Candidate model       | ‑0.29 | [‑0.6, ‑0.06]  | 0.16      |
> | Alignment‑induced representational shift | Data‑generating model | ‑0.25 | [‑0.61, ‑0.12] | 0.004     |
> | Alignment‑induced representational shift | Candidate model       | 0.52  | [0.33, 0.88]   | 0.0001    |
>
> *(ID: GRIDE intrinsic dimensionality estimate)*
>
> Intrinsic dimensionality—whether measured on original or transformed features—was not a significant predictor of the model recovery outcome (p > 0.15), whereas the alignment-induced representational shift remains the dominant, statistically significant predictor. Note that our significance threshold accounts for multiple comparisons, making the test relatively stringent.
>
> We will include the updated GRIDE-based results as a supplemental figure in the revised manuscript.
>
> **3. Broader impact.** To avoid redundancy, we kindly refer the reviewer to our response to the second point of reviewer Suzu. In brief, we believe that while identifying models that mechanistically explain human brain and behavior is a central ambition in neuroAI and cognitive computational neuroscience, current benchmarking practices tend to reward predictive performance alone—often without asking whether the evaluation metrics can reliably identify the correct model. Our contributions are threefold: (1) we characterize the severity of this issue in an important benchmark; (2) we introduce a general approach for evaluating model recovery accuracy using noise-calibrated simulations; and (3) we aim to raise awareness of the tradeoff between predictivity and identifiability, advocating for model recovery simulation as a standard component of the analytic pipeline when comparing computational models.
>
>
> **4. Misidentification due to stochastic optimization and $W$ matrix initialization.** In all experiments, we optimized the matrix $W$ using full-batch L-BFGS and initialized $W$ to the identity matrix for both the data-generating and candidate models. Given fixed data, initialization, and optimizer hyper-parameters, full-batch L-BFGS is deterministic; consequently, the optimizer does not introduce stochasticity. The only stochastic elements in our pipeline are the triplet-sampling procedure described in Appendix A.3, and the sampling of the simulated responses from the discrete probability distribution defined by the data-generating model. The random seed was varied across repeats of the same experimental condition.
>
> **6. Human test-retest reliability.** We agree with the reviewer that studying inter- and intra-subject response variability would be an insightful direction for future work, and we are actively looking into this question. The current noise ceiling estimate reflects both sources of variability.
>
> We believe the reviewer would agree that model recovery accuracy and inter- and intra-subject response reliability are related yet operate on different scales: even if the behavioral responses are unreliable (e.g., 50%), given many such responses, model recovery accuracy can still approach 100%. For example, consider comparing two biased coins, one with $p=0.5$ and the other with $p=0.55$. A single coin toss is not very informative, but many independent tosses can reliably reveal which of the two coins we are holding.

---

> > ### Comment · Reviewer_GT1J · 2025-08-01
> >
> > I thank the authors for their thorough response. I’m convinced of the broader implications of their great work, and will therefore raise my score to 6.

---

### Official Review · Reviewer_SokD · 2025-07-02

**Clarity:** 3
**Significance:** 2
**Originality:** 2
**Rating:** 4
**Confidence:** 5

**Summary:**

The authors conduct large-scale model-recovery simulations on the THINGS odd-one-out behavioral dataset to show that linear probing of DNN representations (essentially flexible evaluations that allow linear reparamerizations of the feature space) fails to reliably recover the true data-generating model. Things improve with more samples but the identifiability of the true model is still below 80% with millions of trials. They show a trade-off between predictivity of human behavior and identifiability of the true model as the flexibility of the probe increases (going from simple scaling to a rectangular to a full linear transformation matrix). The authors also try to identify factors causing misidentifiability and find that representations with higher effective dimensionality and those that incur higher geometry shifts after alignment with behavioral responses are the ones that are more misidentified.

**Questions:**

My main concern is about the generality of findings (that more flexible mappings reduce identifiiability) beyond behavioral tasks. All simulations use the THINGS odd-one-out behavioral paradigm. Yet the paper’s rhetoric (“flexible evaluation may incur a hidden cost”) implies broad applicability to any linear alignment metric—most pressingly those applied to neural data . It is hard to know if the finding obtained with one dataset would generalize to broader settings, particularly neural data. Without at least a toy simulation using a realistic neural noise model, it’s unclear whether the same identifiability collapse holds, improves, or worsens in neural-alignment settings.

The authors should also link their finding that flexible evaluations reduce identifiability with other findings in neuroscience. for eg that of conwell et al which show that when flexible evaluations are employed (eg linear predictivity), many different candidate models achieve similar brain predictivity. So in some sense, this issue has already been flagged in the field.

The paper could also benefit from discussion of other potential criteria that should be used for evaluating methods of identifying candidate models of brain/behavior. Identifiability is an important (almost necessary one) but are there other criteria that we want our evaluation methods to satisfy? e.g., interpretability of weights, robustness to noise etc

The experiments sweep from zero-shot through diagonal, low-rank, and full linear transforms—but do not examine any class of nonlinear mapping models. It would be valuable to see whether moderately nonlinear mappingsfurther amplify or mitigate identifiability failures. Beyond diagonal/low-rank/full, it would also have been useful to introduce other structured‐linear probes (e.g. sparse at different sparsity levels) and chart their place on the accuracy–identifiability curve. The current probe evaluation (diagonal vs low-rank vs full seems too limited)

Also the paper flags the main issue of flexible models as their misidentification rate. But which studies use model-behavior comparisons for predictivity vs using the most human behavior like model  to make conclusions about what shapes human behavior.  It would help to cite a few examples of such articles whose inference pipelines depend on the reliability and identifiability of flexible linear probes.

Overall, this paper issues an important warning about over-flexible linear probes in behavioral model comparison, and suggests that similar pitfalls likely afflict neural-alignment studies. However, to substantiate its broader implications and to offer concrete remedies, it needs (1) at least a small neural-data simulation and (2) a thorough exploration of structured and mildly nonlinear probes.

**Ethical Concerns:**

["NO or VERY MINOR ethics concerns only"]

**Final Justification:**

I have increased my score to borderline accept. I still don't know to what extent the conclusions from this study apply to model-brain comparisons (which was cited as a key motivation behind this paper).

**Limitations:**

yes

**Quality:**

3

**Strengths And Weaknesses:**

The results are intuitive and the paper has a nice take-home that also seems actionable: that whenever you use highly flexible, data-driven alignment (linear or otherwise) to compare models to brain or behavior, you risk mistaking best fit for true fit (predictability vs identifiability trade-off).
The paper is well written, the experiments are sound and the conclusions are straightforward.

---

> ### Author Rebuttal · Authors · 2025-07-31
>
> We thank the reviewer for their thoughtful response and deep engagement with the manuscript.
>
> **Generalization to neural data.** We agree that generalization to neural data is non-trivial. Model identification using neural data operates in a markedly different regime: responses are multivariate and continuous rather than univariate and discrete.
>
> Following the comments of SokD and other reviewers, we carefully considered the possibility of conducting a simulation of model recovery using neural data with a *realistic neural noise model*. We concluded that this would be a substantial undertaking. A realistic simulation would entail accurately capturing both single-trial variability and inter-channel noise correlations. For the NSD fMRI dataset, a complete reanalysis would be required because GLMSingle does not provide statistically independent single-trial response estimates (across-trial reliability is used to guide HRF choice and signal denoising). For MUA or spike-rate data, access to single-trial responses is easier, but modeling must account for the multivariate, non-Gaussian distribution of the noise (i.e., faithfully capturing response variability and noise correlations). Furthermore, a realistic simulation may require modeling modality-specific noise sources, such as head motion in fMRI, or slow drift in response amplitudes in electrophysiology.
>
> Hence, we believe that such a study would justify a separate paper devoted to this problem. Combining Things-Odd-One-Out and one or more neural datasets into a single paper would considerably broaden the scope. However, it would not allow for sufficient detail and depth in describing the various analytic choices required for each simulation study. Here, we chose to focus on a single but important test case and to pursue it in as much depth as possible. We hope we can gain the reviewer's support for publishing the present work in NeurIPS, despite our deliberate focus on a behavioral test case at this stage.
>
> We have revised our draft to reflect the uncertainty about the generalization to neural data. Revised limitations paragraph:
> > The scope of our simulations is limited to behavioral data, and specifically, to the THINGS odd-one-out task. We chose this task as a test case because it is supported by a large empirical dataset (Hebart et al., 2023) and allows straightforward simulation of synthetic responses by sampling from model-specified multinomial distributions. The noise-calibrated simulation approach can be readily extended to other behavioral paradigms, such as classification (Battleday et al., 2020) or multi-arrangement (Kriegeskorte \& Mur, 2012). Model identification using neural data operates in a markedly different regime: responses are multivariate and continuous, rather than univariate and discrete as in the behavioral case. Therefore, while our results demonstrate a pronounced predictivity–identifiability trade-off when comparing models to behavior in a large dataset, the severity of this trade-off for neural data cannot be inferred from our findings.
>
> We also added the word "behavioral" to the abstract's conclusion:
> > These findings demonstrate that, even with massive behavioral data, overly flexible alignment metrics may fail to guide us toward artificial representations that are genuinely more human-aligned.
>
> We wish to keep the sentence "flexible evaluation may incur a hidden cost". It presents our working hypothesis rather than the paper's empirical conclusion.
>
> **Link to recent neuroscientific reports on flexible evaluation.** We agree that these links are important, and in part, they have motivated our work. The remainder of our new limitations paragraph spells this out.
>
> > Recent reports of qualitatively distinct neural network models achieving indistinguishable performance under flexible comparisons to neural data (Conwell et al., 2024; Storrs et al., 2021) make this question especially pertinent. Addressing it will require future work using noise-calibrated, modality-specific neural simulations.
>
> Note that the equivalent model performance reported in Storrs et al. and Conwell et al. does not prove that the compared models make equivalent predictions. Similar accuracy is a necessary but insufficient condition for prediction equivalence. For example, models with similar aggregate performance may produce different error patterns (see Rajalingham et al., 2018, J. Neurosci.) or explain different subsets of the data. Therefore, comparing each model to neural or behavioral data does not eliminate the need to study how distinguishable the predictions of different models are, as we do here.
>
> **Additional criteria for evaluation measures.**
>
> This is a good point. Given the limited space, we incorporated it by expanding the second future direction in the discussion:
> > Constraining data-driven model alignment by biologically motivated and/or inter-individual variability–informed priors (...) may improve upon the overly flexible family of linear transformations. Furthermore, imposing greater constraints on the readout may enhance its interpretability. For example, constraining the learned stimulus embeddings to be non-negative prevents features from canceling each other (e.g., Mahner et al., 2025). Finally, Bayesian readout models, which estimate a distribution of feature weights rather than a point estimate, may improve robustness to sampling noise.
>
> **Nonlinear and structured linear probes.** We would be happy to introduce additional probes into our analysis. We will update the next version of the manuscript regardless of whether it is a camera-ready version or a resubmission. We have already introduced a more traditional Frobenius-norm penalty on the weights and found that it is slightly less effective for model identification than the penalty introduced in the paper.
>
> However, we would be grateful for clarification on the suggested probes. Could the reviewer clarify what kind of nonlinear probes they envision for this task? The vast majority of the literature purposely relies on linear probes to ensure interpretability. Regarding structured linear probes, since we are modeling higher-order perceptual judgments, penultimate layers are used (following Muttenthaler et al., ICLR 2023). This rules out separable readout (i.e., distinct weights for feature channels and spatial locations). The remaining probes seem to be sparse. Does the reviewer have in mind probes that are sparse in the linear transformation matrix, or probes that are sparse in the resulting embedding matrix (as in the SPoSE and VICE embedding models)? If it is the former, are there any examples in the literature where sparse readout is motivated and implemented?
>
> **Distinguishing Predictive Performance from Model Identification.** All of the human–DNN comparison studies cited in the manuscript are aimed at understanding the brain, rather than predicting responses as an end in itself. However, we agree that—especially in the context of NeurIPS—it is worthwhile to explicitly distinguish between prediction for its own sake and prediction used as a means for model identification. We have edited the introduction to make this distinction and to cite several studies that use flexible evaluation of behavioral model predictions, with the aim of understanding the underlying biological computations:
>
> > When evaluation is made flexible by fitting linear weights to improve the alignment of neural network representations with brain [...] or behavioral data (e.g., Peterson et al., 2018, Cognitive Science; Battleday et al., 2020, Nature Communications; Daube et al., 2021, Patterns; Muttenthaler et al., 2022, ICLR), this approach achieves predictive accuracy that exceeds that of any other computational model.
> In some neuroscientific applications (e.g., brain--computer interfaces), accurate prediction is useful regardless of the underlying mechanism. In contrast, basic-science studies in computational neuroscience often rely on the assumption that a neural network whose representations are more predictive of brain or behavioral data is a better model of the mechanisms underlying the biological data.

---

> > ### Comment · Reviewer_SokD · 2025-08-05
> >
> > I thank the authors for the through response. I appreciate the clarifications that the authors plan to include in the revised manuscript and have raised my score in light of this. However, I'm still unsure about the generalization of these findings to the neural domain (given that the paper motivates their analyses based on model-brain comparisons) but I understand the challenges in extending this to neural data. For other probes, I was suggesting employing a shallow ANN as a nonlinear probe and linear probes (with l1 sparsity at different alphas) to probe more densely how mapping complexity affects conclusions.

---

> > > ### Author Response · Authors · 2025-08-06
> > >
> > > We fully share your interest in extending the noise-calibrated recovery analysis to fMRI and electrophysiology. Because a faithful simulation requires developing a modality-specific noise model and calibration to specific datasets, we believe a standalone study would better serve the community than an appendix. We have clarified in the manuscript the precise scope and limitations of our results, with an emphasis on the fact that generalization to neural data is non-trivial.
> > >
> > > Regarding probe selection, our study determines regularization strength via nested cross-validation rather than fixing the penalty hyperparameter a priori. This procedure, grounded in current ML methodology and the work of Muttenthaler et al. (2023), yields a realistic best-case estimate of model performance. We would be pleased to apply the same nested-CV strategy to (i) an ℓ₁-penalized linear probe and (ii) a shallow ANN probe, and to report our findings in the predictive-accuracy-model-identifiability trade-off section. Would this address your concern adequately?
> > >
> > > Thank you again for your constructive feedback.

---

### Official Review · Reviewer_Suzu · 2025-07-03

**Clarity:** 3
**Significance:** 3
**Originality:** 2
**Rating:** 5
**Confidence:** 2

**Summary:**

This paper investigates whether linear probing can reliably identify which model generated behavioral data in human perception tasks. The authors conduct large-scale model recovery simulations using 20 diverse vision models on the THINGS odd-one-out dataset. They find that even with massive datasets, model recovery accuracy plateaus below 80%, meaning the best-fitting model often isn't the data-generating model.

**Questions:**

Your study only looks at odd-one-out similarity judgments, which is pretty narrow. Do you think these problems with model recovery would show up in other kinds of behavioral tasks or even neural data? It would be helpful to see some theoretical reasoning or even small pilot studies showing this isn't just a quirk of this particular dataset. I'd say this is my primary qualm with the manuscript at this moment.

**Ethical Concerns:**

["NO or VERY MINOR ethics concerns only"]

**Final Justification:**

The authors' response, especially the new set of results, address some of my concerns sufficiently so I raised my score.

**Limitations:**

yes

**Paper Formatting Concerns:**

- Some figures (especially confusion matrices) are difficult to interpret due to dense labeling. I'd suggest de-cluttering the figures.

**Quality:**

3

**Strengths And Weaknesses:**

Strengths
- Very thorough evaluation in terms of testing 20 diverse models across multiple dataset sizes (50 to 4.2M trials), with noise calibration to match human response variability.
- The regression analysis identifying specific factors (representational geometry shifts, effective dimensionality) that drive misidentification provides useful mechanistic insights.
- I really enjoyed the discussion section “Navigating the accuracy–identifiability tradeoff” at the end of the paper.

Weaknesses
- The study is restricted to one behavioral task of odd-one-out judgments and one dataset. The quantitative results are specific to this particular model set and may not generalize broadly. The paper does draw attention to this in its limitation section, but still, for a venue like NeurIPS, the scope feels narrow.
- The core insight that flexible metrics can overfit isn't entirely novel; this has been recognized in various forms across ML. The specific quantification in this behavioral context is useful but somewhat incremental.
- I wonder if this work would be better suited for a more specialized venue focused on computational cognitive science or model evaluation methodologies rather than NeurIPS.

---

> ### Author Rebuttal · Authors · 2025-07-30
>
> We thank the reviewer for their positive evaluation and detailed review.
>
> **In response to:** *“The study is restricted to one behavioral task of odd-one-out judgments and one dataset. The quantitative results are specific to this particular model set and may not generalize broadly.”*
>
> We selected the THINGS odd-one-out paradigm because its massive scale (4.7 million human judgments) provides favorable conditions for identifiability, and its simple three-way response paradigm allows precise probabilistic simulation. Therefore, we believe that model recovery accuracy plateauing below 80% even with millions of simulated trials indicates a fundamental limitation of flexible linear evaluation, rather than a quirk of this dataset.
>
> We agree that the particular composition of the model set may influence recovery outcomes. To test this possibility, we conducted additional simulations after the submission deadline, using an extended model set. This set included the 20 models reported in the manuscript along with OpenCLIP ConvNext XL, ALIGN, FLAVA Full, SigLIP2 B, BLIP 2, EVA02 CLIP-enormous, PE Huge, ImageBind Huge, OpenAI CLIP ViT Large, and AlexNet. When curating the new models, we prioritized multi-modal (visual-language) models, motivated by the strong performance of CLIP in the original evaluation.
>
> Below, we report the model recovery performance of the extended model set as a function of dataset size:
>
> | Number of training triplets | Model recovery accuracy | 95% CI      |
> | --------------------------- | ----------------------- | ----------- |
> | 100                         | 0.07                    | [0.04, 0.10] |
> | 400                         | 0.11                    | [0.08, 0.15] |
> | 1,600                       | 0.18                    | [0.13, 0.22] |
> | 6,400                       | 0.26                    | [0.21, 0.31] |
> | 25,600                      | 0.36                    | [0.31, 0.41] |
> | 51,200                      | 0.43                    | [0.37, 0.48] |
> | 204,800                     | 0.62                    | [0.56, 0.67] |
> | 819,200                     | 0.66                    | [0.61, 0.72] |
> | 1,638,400                   | 0.70                    | [0.64, 0.75] |
> | 4,200,000                   | 0.70                    | [0.65, 0.75] |
>
> Notably, OpenAI CLIP ConvNeXT XL was not consistently identified when it was the data-generating model. We will incorporate these simulations along with a detailed confusion matrix in the revised (camera-ready or resubmitted) version of the paper. Overall, the results indicate that model confusability is not limited to the original 20-model set reported in the current version of the manuscript.
>
> **In response to:** *The core insight that flexible metrics can overfit isn't entirely novel; this has been recognized in various forms across ML. The specific quantification in this behavioral context is useful but somewhat incremental. I wonder if this work would be better suited for a more specialized venue focused on computational cognitive science or model evaluation methodologies rather than NeurIPS.*
>
> We agree with the reviewer that the insight that flexible evaluation can lead to overfitting is well established in statistics and machine learning. It is also increasingly acknowledged in neuroAI and cognitive computational neuroscience. However, most neuroscientific studies comparing DNN and neural/behavioral responses place strong emphasis on maximizing predictivity, and seldom evaluate how well their chosen alignment metrics support model identification. Here, we analyze model identification performance in a leading benchmark—the largest currently available behavioral dataset—using a realistic simulation and a continuum of relevant evaluation metrics. To our knowledge, no prior work has systematically quantified model identification performance under realistic, noise-calibrated conditions. While our analysis focuses on a specific benchmark, the methodology we introduce and the substantial identifiability–predictivity trade-off it uncovers are relevant to a broader class of model-evaluation problems. We hope that this work will encourage researchers to conduct such analyses as an integral part of their empirical studies.
>
> We agree that this manuscript could also be suitable for specialized venues such as CCN or NBDT. However, many researchers active in neural network modeling of brain and behavior are more likely to be exposed to a NeurIPS paper. Several studies that rely on flexible metrics for model identification have themselves been published at NeurIPS/ICLR/ICML. Publishing this work at NeurIPS can help ensure that the identifiability–predictivity trade-off is recognized and addressed by the broader modeling community.
>
> **In response to:** *“Your study only looks at odd-one-out similarity judgments, which is pretty narrow. Do you think these problems with model recovery would show up in other kinds of behavioral tasks or even neural data?"*
>
> As discussed in our response to the reviewer's first point, we do not see a reason that other behavioral tasks would yield better model identification performance. Moreover, existing real behavioral datasets are at least an order of magnitude smaller than THINGS-odd-one-out, and therefore even more prone to model misidentification.
>
> As for neural datasets, the statistical regime is different—the responses are multivariate and continuous and rather than univariate and discrete. We are therefore more cautious about making strong predictions about these settings without additional data. Please see our response to Reviewer SoKD below for a more detailed discussion of this point, as well as a relevant modification of the manuscript.

---

### Official Review · Reviewer_jQvh · 2025-07-03

**Clarity:** 3
**Significance:** 4
**Originality:** 4
**Rating:** 6
**Confidence:** 4

**Summary:**

The authors address the hypothesis that good brain data predictivity in neural network models is reflective of a corresponding architectural similarity with the entity that generated said brain data, i.e. that being good at predicting brain data reflects mechanistic alignment between models and human subjects. Standard assumption is that this is the case, but drawing conclusions from such approaches is made hard by the presence of noise in the measurements and the limited data available for model identification. The authors study whether model identification is possible even when data availability is not an issue and the data generating procedure is known. They do this by substituting human participants with a neural network, recording its responses and checking if model identification can pick out the data generating architecture from a set of candidates at evaluation time. They find that in general, even with millions of examples, such identification is not possible. These results cast doubt on the possibility of identifying the mechanisms used by humans to represent data using brain-predictivity methods alone.

**Questions:**

I have no interesting questions at this point. If some come up I will post them.

**Ethical Concerns:**

["NO or VERY MINOR ethics concerns only"]

**Final Justification:**

I believe this work provides a good way to think about how representation alignment as currently computed/formulated may not match our intuitions of how it should works. This is important as, in my opinion, this field is riddled with misconceptions, concept creep, and straight-up motte-and-baileying of claims. Works such as help illustrate with clear experiments how incorrect assumptions can lead to false conclusions. I am also happy with the authors replies and while my score won't increase, I definitely recommend it being accepted.

**Limitations:**

The authors clearly state the limitations of the study, but I think what they have done is more than sufficient.

**Paper Formatting Concerns:**

None.

**Quality:**

4

**Strengths And Weaknesses:**

Strengths:
- The paper is well motivated, addressing a key question that has generated significant debate recently.
- The method is a clever but simple way to side-step the issue of comparing to humans, by just focusing on whether models can identify each other first, which is a much simpler task.
- The methods are well detailed, explaining how the experimental setting is matched as much as possible to what we would encounter when comparing to humans.


Weaknesses:
- The study is restricted to behavioral data, though I agree with the authors that this is a good starting point.
- The explanation on how the noise ceiling was determined could be a bit more clear.

---

> ### Author Rebuttal · Authors · 2025-07-31
>
> We thank the reviewer for their thoughtful and supportive evaluation of our work. Below we address each point in turn.
>
> **1. Limitation to behavioral data**
> We appreciate the reviewer's endorsement of behavioral data as an appropriate scope for a first noise-calibrated simulation study of this problem, and we intend to extend this work to neural data in follow-up research. We refer the reviewer to our reply to Reviewer SoKD for more details on how we understand the relationship between the behavioral and neural domains in this context.
>
> **2. Noise ceiling estimation details.** We thank the reviewer for the suggestion to make this description clearer. To estimate the noise ceiling, the THINGS odd-one-out dataset includes a subset of 1,000 triplets—each presented to approximately 30 participants. To compute a lower-bound on the performance of the ideal model, we used a leave-one-subject-out procedure on this data, following Hebart et al., 2020 (Nature Human Behaviour). For each triplet and each participant, we withheld that participant’s response, determined the majority choice among the remaining responses, and recorded whether the held-out response matched the majority. We repeated this for all participants, averaged the match rates across triplets. This reproduced the noise-ceiling values reported by Hebart et al. We have added the following paragraph under “Calibration” in the Methods section:
> >We estimated the noise ceiling with a leave-one-subject-out procedure: for every triplet, we removed one participant’s response, took the majority vote of the remaining responses, and noted whether the held-out answer matched that vote. We repeated this step for all participants and averaged the match rates across all triplets.

---

> > ### Comment · Reviewer_jQvh · 2025-08-05
> >
> > I thank the reviewers for their reply. Great work.

---

### Official Review · Reviewer_LzV6 · 2025-07-04

**Clarity:** 3
**Significance:** 3
**Originality:** 3
**Rating:** 5
**Confidence:** 4

**Summary:**

The paper questions the common assumption that high predictive accuracy of human behavior is a good proxy for internal representational alignment. To test this, the authors take 20 pre-trained vision models, freeze their weights, and learn a linear transformation of each model's features to best predict judgments from the THINGS dataset. These aligned models, which could be called "teacher" models, then generate synthetic datasets. In the recovery phase, the same 20 frozen models ("student" models) are fitted to the synthetic data by learning a new linear transformation. The authors then tests if the original data-generating model can be correctly identified based on predictive accuracy.

The authors find that while model recovery improves with dataset size, it plateaus below 80% accuracy. The errors are systematic, with certain models often being misidentified as the best predictor. They find that that greater flexibility in the alignment metric improves predictive accuracy but hurts model identifiability.

**Questions:**

Recent work (e.g., Lampinen et al., 2024) has focused on how training objectives create distinct representational biases. How do you see these biases factoring into your analysis? It seems they would determine the initial representational geometries that your linear probe then attempts to align.

Your work shows that DNNs can be confused for other DNNs. How do you think these claims extend to the much harder problem of comparing DNNs to the brain? It seems your findings represent a best-case scenario -- the problem of misidentification may be even more severe in the neuroscience setting.

**Ethical Concerns:**

["NO or VERY MINOR ethics concerns only"]

**Final Justification:**

I have considered the discussion with the authors and other reviewers. My concerns have largely been addressed and I believe the paper is strong. I will maintain my positive rating.

**Limitations:**

Yes.

**Paper Formatting Concerns:**

No.

**Quality:**

3

**Strengths And Weaknesses:**

**Strengths**

The model recovery setup is novel, as it allows for evaluation against a known ground truth.

The approach is applied to a large set of 20 vision architectures and, calibrates the synthetic data to realistic human noise levels, making the setup more realistic. The authors go beyond simply identifying the problem and use regression analysis to find predictors of misidentification.

**Weaknesses / Feedback**

Some of the presentation was difficult to parse. Adopting the "teacher/student" model analogy, as the setup is similar to distillation, would make the simulation process much more intuitive for readers.

The conclusion could benefit from proposed avenues of research to address the issue the paper identifies.

---

> ### Author Rebuttal · Authors · 2025-07-31
>
> We thank the reviewer for their interest in our manuscript and for the constructive review.
>
> **1. Clarifying the model-recovery simulation using a student–teacher analogy.** We appreciate the suggestion and have incorporated this analogy into our Methods section:
>
> > This setup is analogous to data distillation: the candidate models act as "students" attempting to approximate the input–output function of a "teacher"—the data-generating model. Model recovery is successful when the best-performing student is the teacher itself, rather than an alternative model.
>
> **2. Elaborating on proposed avenues to address the issue identified in the paper.** We outlined several possible research directions at the end of the Discussion section, under the heading "Navigating the accuracy–identifiability tradeoff." In response to this and related comments, we have revised and expanded the section, further elaborating on possible approaches to mitigate the issue identified in our work.
> >*1. Change the stimuli:* As in many model comparison studies, we evaluated models out-of-sample, that is, on new stimuli drawn from the training distribution. Out-of-distribution generalization, which more strongly probes the models' inductive biases, may offer greater model comparative power. Stimuli designed to elicit model disagreement may yield even greater gains.
> The recovery gains we obtained from larger and more diagnostic triplet sets suggest that smarter sampling matters at least as much as sheer volume (Fig. 1). Adaptive, model-driven stimulus selection constructing trials that maximize the expected divergence in network responses—can sharpen our ability to treat predictive accuracy as an indicator for human—model alignment, enhancing current flexible alignment methods without compromising identifiability.
>
> >*2. Change the metrics:* Constraining data-driven model alignment by biologically motivated and/or inter-individual variability–informed priors may improve upon the overly flexible family of linear transformations. Furthermore, imposing greater constraints on the readout may enhance its interpretability. For example, constraining the learned stimulus embeddings to be non-negative prevents features from canceling each other (e.g., Mahner et al., 2025). Finally, Bayesian readout models, which estimate a distribution of feature weights rather than a point estimate, may improve robustness to sampling noise.
>
> >*3. Change the models:* The considerable geometric shifts required to align the networks suggest that linear probes can obscure important representational mismatches. Embedding richer priors directly into the models—through task design, objective functions, or biologically inspired architectures—could allow aligned representations to emerge natively, reducing the need for substantial post hoc transformations. More broadly, further progress in neural network-based modeling of brain and behavior may depend less on ever-larger data-driven fits of pre-trained models and more on deliberate model refinement to embody explicit computational hypotheses.
>
> **3. The effect of training objectives on model recovery accuracy.** We thank the reviewer for highlighting this for our consideration. After the submission deadline, we introduced ten additional models, primarily image–text-aligned (e.g., ALIGN, OpenCLIP-ConvNeXt-L, OpenAI-CLIP ViT-L) and reran the model-recovery simulations. To address this specific comment, we estimated a between-objective confusion matrix, in which model recovery is considered incorrect only if it confuses models trained on different objectives. However, even under this permissive criterion, model recovery plateaued at 72%, compared to 70% under the standard criterion applied to the same model set. We have added a summary of this result to the Results section of the manuscript:
>
> >**Model-recovery performance plateaus despite objective-driven representational divergence**
> Recent work (Lampinen et al., 2024) has shown that different training objectives produce distinct representational biases. We categorized each model as supervised, unsupervised, or image–text-aligned and then performed a between-objective model-recovery analysis. Even with a 4.2-million-triplet training set, objective-based grouping yielded only a modest accuracy gain: from approximately 70% to 72%. These results indicate that, although initial internal representations reflect objective-specific biases (Lampinen et al., 2024), flexible evaluation largely obscures these differences.
>
> 4.**Extension beyond behavioral settings.** We expect that the predictivity–identifiability trade-off will likewise manifest in neural data. However, the severity of this trade-off may change, because model comparison using neural data operates in a somewhat different regime: responses are multivariate and continuous, rather than univariate and discrete. We have revised the discussion section to reflect this uncertainty:
> > The scope of our simulations is limited to behavioral data, and specifically, to the THINGS odd-one-out task. We chose this task as a test case because it is supported by a large empirical dataset (Hebart et al., 2023) and allows straightforward simulation of synthetic responses by sampling from model-specified multinomial distributions. The noise-calibrated simulation approach can be readily extended to other behavioral paradigms, such as classification (Battleday et al., 2020) or multi-arrangement (Kriegeskorte \& Mur, 2012). Model identification using neural data operates in a markedly different regime: responses are multivariate and continuous, rather than univariate and discrete as in the behavioral case. Therefore, while our results demonstrate a pronounced predictivity–identifiability trade-off when comparing models to behavior in a large dataset, the severity of this trade-off for neural data remains an open question.
>
> We aim to resolve this uncertainty in follow-up work. To reduce the risk of misinterpreting the scope of our empirical results, we added the word “behavioral” to the concluding sentence of the abstract:
> > These findings demonstrate that, even with massive behavioral data, overly flexible alignment metrics may fail to guide us toward artificial representations that are genuinely more human-aligned.

---

> > ### Comment · Reviewer_LzV6 · 2025-08-06
> >
> > I thank the authors for their thoughtful response to my comments and updates to the paper. I enjoyed the paper!

---

### Note · Authors · 2025-08-15

We are grateful to the reviewers for their thorough and constructive engagement with our work. Your feedback has improved the manuscript’s clarity and precision. We have updated it to more clearly delineate its behavioral scope, broaden the candidate model set, and include intrinsic dimensionality measures. Additional analyses suggested by the reviewers are underway and will be included in the revised or camera-ready version.

---

### Decision · Program_Chairs · 2025-09-17

**Decision:**

Accept (poster)

**Comment:**

This submission evaluates representational similarity measures using an odd-one-out task paradigm. While reviewers were generally positive about the thoroughness of the work and the novelty of applying the odd-one-out validation approach, I do have some concerns about the positioning of the contributions relative to existing literature.

The idea of validating representational similarity through model recovery has become standard practice across neuroscience and machine learning communities. Notable prior work includes:

* CKA fails to recover same-architecture inits ([Han et al., 2023](https://proceedings.mlr.press/v202/han23d))
* RSA recovers ground-truth geometries ([Schütt et al., 2023](https://elifesciences.org/articles/82566))
* RSA etc. fail to achieve high sensitivity/specificity in recovery of functional similarity i.a. ([Klabunde et al., 2023](https://arxiv.org/abs/2305.06329); [Ding et al., 2021](https://arxiv.org/abs/2108.01661); [Bo et al., 2024](https://arxiv.org/abs/2411.14633))
* And the classic [Jonas & Kording (2017)](https://doi.org/10.1371/journal.pcbi.1005268)

While I acknowledge that the odd-one-out task paradigm has not previously been used to directly validate RSA, etc. for model recovery, and this represents an interesting methodological contribution, the broader findings about failures of representational similarity align with established knowledge in the field.

Nevertheless, given the thorough execution and the new methodological contribution of using odd-one-out specifically, I recommend acceptance contingent on a minor revision that:

1. More clearly articulates the specific contributions relative to the prior work cited above;
2. Narrows the scope of claims in the paper (including potentially revising the title) to better reflect what is genuinely new; and
3. Highlights the specific insights that the odd-one-out validation paradigm provides beyond existing model recovery approaches.

These revisions would help readers understand precisely how this work advances our understanding of representational similarity metrics beyond documenting failures that have been previously established.